# Ciliary length regulation by intraflagellar transport in zebrafish

**Yi Sun[1,2†], Zhe Chen[1,3†], Minjun Jin[1,2], Haibo Xie[1,2]\*, Chengtian Zhao[1,2,4]\***

[1]Institute of Evolution & Marine Biodiversity, Ocean University of China, Qingdao, China; [2]Fang Zongxi Center for Marine Evo Devo, MOE Key Laboratory of Marine Genetics and Breeding, College of Marine Life Sciences, Ocean University of China, Qingdao, China; [3]Tsinghua-Peking Center for Life Sciences, Beijing Frontier Research Center for Biological Structure, McGovern Institute for Brain Research, State Key Laboratory of Membrane Biology, School of Life Sciences and MOE Key Laboratory for Protein Science, Tsinghua University, Beijing, China; [4]Laboratory for Marine Biology and Biotechnology, Qingdao Marine Science and Technology Center, Qingdao, China

**\*For correspondence:**
xiehaibo@ouc.edu.cn (HX);
chengtian_zhao@ouc.edu.cn (CZ)

[†]These authors contributed equally to this work

**Competing interest:** The authors declare that no competing interests exist.

## eLife Assessment

The manuscript represents a **valuable** conceptual and technical contribution to our understanding of ciliogenesis and intraflagellar transport in vertebrates. Through a series of **solid** and technically superb live imaging experiments to directly visualize intraflagellar transport in various zebrafish ciliated tissues, the authors unveil the surprising breadth of intraflagellar transport speed among differing organs and link this to cell type-specific differences in cilia length and intraflagellar transport train size. This work will be of broad interest to researchers in numerous fields, including development, cell biology, and imaging.

**Abstract** How cells regulate the size of their organelles remains a fundamental question in cell biology. Cilia, with their simple structure and surface localization, provide an ideal model for investigating organelle size control. However, most studies on cilia length regulation are primarily performed on several single-celled organisms. In contrast, the mechanism of length regulation in cilia across diverse cell types within multicellular organisms remains a mystery. Similar to humans, zebrafish contain diverse types of cilia with variable lengths. Taking advantage of the transparency of zebrafish embryos, we conducted a comprehensive investigation into intraflagellar transport (IFT), an essential process for ciliogenesis. By generating a transgenic line carrying Ift88-GFP transgene, we observed IFT in multiple types of cilia with varying lengths. Remarkably, cilia exhibited variable IFT speeds in different cell types, with longer cilia exhibiting faster IFT speeds. This increased IFT speed in longer cilia is likely not due to changes in common factors that regulate IFT, such as motor selection, BBSome proteins, or tubulin modification. Interestingly, longer cilia in the ear cristae tend to form larger IFT compared to shorter spinal cord cilia. Reducing the size of IFT particles by knocking down Ift88 slowed IFT speed and resulted in the formation of shorter cilia. Our study proposes an intriguing model of cilia length regulation via controlling IFT speed through the modulation of the size of the IFT complex. This discovery may provide further insights into our understanding of how organelle size is regulated in higher vertebrates.

## Introduction

Understanding how cells regulate the size of their organelles is a fundamental question in cell biology. However, the three-dimensional complexity of most organelles poses challenges for accurate measurement, and thus, the underlying regulation mechanisms remain largely elusive. In contrast to

the localization of most organelles within cells, cilia are microtubule-based structures that extend from the cell surface, facilitating a more straightforward quantification of their size. Cilia are highly conserved organelles found across various organisms, ranging from protozoa to humans. Their formation relies on microtubules, and their size can be easily measured by their length, making cilia an ideal model for studying size regulation of sub-cellular organelles within the same organisms (*Chan and Marshall, 2012*).

Cilia play a crucial role in regulating various physiological and biochemical processes (*Goetz and Anderson, 2010*; *Nachury and Mick, 2019*; *Singla and Reiter, 2006*). Structural or functional abnormalities of this organelle can result in a wide range of human genetic disorders, including retinal degeneration, polycystic kidneys, and mental retardation (*Mill et al., 2023*; *Reiter and Leroux, 2017*; *Song et al., 2016*). The assembly and maintenance of cilia rely on an elaborate process known as intraflagellar transport (IFT; *Kozminski et al., 1993*). The IFT complex is located between the doublet microtubules of the cilia and the ciliary membrane, playing a vital role as a mediator of cargo transportation within cilia (*Bhogaraju et al., 2013*; *Hesketh et al., 2022*; *Meleppattu et al., 2022*). The IFT complex consists of two subcomplexes: the IFT-B complex, which transports the precursors required for cilia assembly from the base to the tip, powered by Kinesin-2; and the IFT-A complex, which transports axonemal turnover products back to the cell body by binding to dynein (*Kardon and Vale, 2009*; *Ou et al., 2007*; *Scholey, 2008*; *Taschner et al., 2012*). Electron microscopy studies have suggested that multiple IFT particles move together along the axoneme, earning the name 'IFT trains' (*Jordan et al., 2018*; *Kozminski et al., 1995*; *Stepanek and Pigino, 2016*). In *Caenorhabditis elegans*, both the slow-speed heterotrimeric kinesin II and the fast-speed homodimeric kinesin OSM-3 (mammalian ortholog, KIF17) coordinate IFT, ensuring the assembly of sensory cilia (*Snow et al., 2004*). The IFT system also interacts with the BBSome complex, dysfunction of which is associated with the human ciliopathy Bardet-Biedl syndrome (*Tsang et al., 2018*). The BBSome complex is moved by IFT and functions to maintain the physical connection between IFT-A and IFT-B subcomplexes in *C. elegans* (*Ou et al., 2005*). Moreover, the BBSome is involved in diverse functions, including protein trafficking into and out of the cilia (*Berbari et al., 2008*; *Jin et al., 2010*; *Lechtreck et al., 2009*; *Liu et al., 2021*; *Nachury et al., 2007*; *Ye et al., 2018*).

The intraflagellar transport system plays a crucial role in ciliogenesis and is highly conserved across various organisms. Interestingly, the length of cilia exhibits significant diversity in different models. For example, *Chlamydomonas* has two flagella with lengths ranging from 10 to 14 μm (*Penny and Dutcher, 2024*), while sensory cilia in *C. elegans* vary from approximately 1.5 μm to 7.5 μm (*Snow et al., 2004*). In most mammalian cells, the primary cilium typically measures between 3 and 10 μm (*Han et al., 2014*; *Li et al., 2021*; *Tu et al., 2023*). Considering the conserved structure of cilia, it becomes intriguing to understand how their length is regulated in different cell types. Extensive research into flagellar length regulation has been conducted in *Chlamydomonas*. After one or two flagella are abscised, the shorter flagella can regenerate to their original length. During this regeneration process, the short flagella undergo a rapid growth phase, transitioning to a slower elongation phase as they approach their steady-state length (*Rosenbaum et al., 1969*). IFT plays a central role in mediating flagellar regeneration and IFT particles usually assemble into 'long' and 'short' trains within the flagella. At the onset of flagellar growth, long trains are prevalent, while the number of short trains gradually increases as the flagellum elongates (*Engel et al., 2009*; *Vannuccini et al., 2016*). Moreover, the rate at which IFT trains enter the flagella, known as the IFT injection rate, is negatively correlated with flagellar length during regeneration (*Engel et al., 2009*; *Ludington et al., 2013*). Researchers have proposed and tested several theoretical models to explain the regulatory mechanism of the IFT injection rate (*Ishikawa et al., 2023*; *Marshall, 2023*; *Wemmer et al., 2020*). Intriguingly, recent studies in the parasitic protist Giardia, which possesses eight flagella of varying lengths, revealed that the ciliary tip localization of microtubule-depolymerizing kinesin-13 is inversely correlated with flagellar length, similar to its role in flagellar length regulation (*McInally et al., 2019*; *Piao et al., 2009*; *Wang et al., 2013*). These findings add another layer of complexity to our understanding of ciliary length regulation and underscore the importance of investigating diverse model organisms to gain comprehensive insights into this fundamental biological process.

Notably, most studies on ciliary length control have focused on single-celled organisms. However, in vertebrates, the presence of highly diverse cell types results in cilia of varying lengths. Understanding how cilia length is regulated in different cell types and why such diversity exists is essential

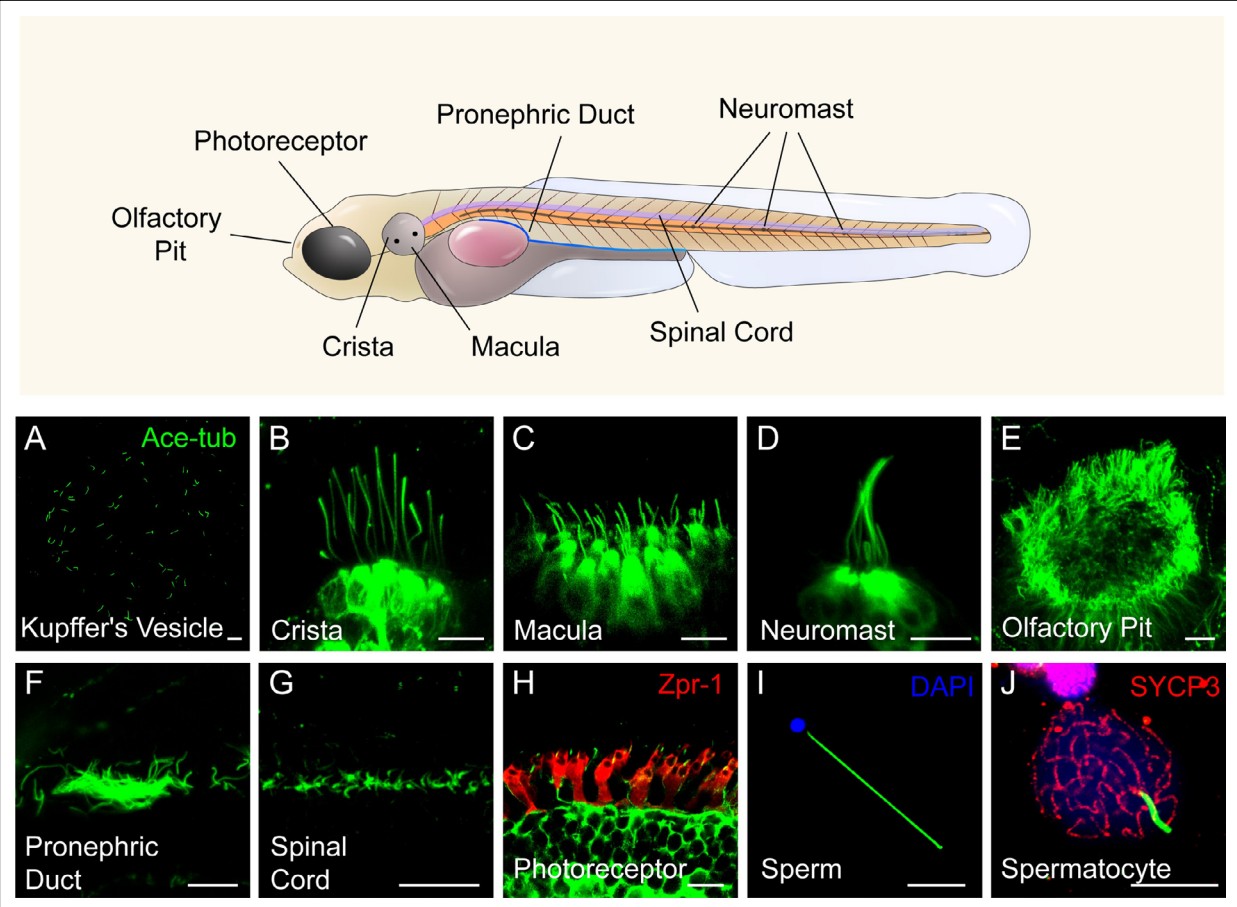

**Figure 1.** Diverse type of cilia are present in zebrafish. (**A**) Cilia in kupffer's vesicle (KV) of a 10-somite stage zebrafish larvae. (**B–H**) Confocal images showing cilia in different type of cells at 4 dpf as indicated. The position of these cells were indicated in the top diagram. (**I–J**) Confocal images showing cilia in the sperm (flagellum) or spermatocyte of adult zebrafish. All the cilia were visualized with anti-acetyleated tubulin. The photoreceptor double cones were stained with zpr-1 antibody in panel H, and nuclei were stained with DAPI in panel I. Immunostaining with anti-SYCP3 labelled the synaptonemal complexes of primary spermatocytes in panel J. Scale bar = 10 μm.

for comprehending the pathogenic mechanisms underlying various organ defects seen in ciliopathies. Unfortunately, direct observation of intraflagellar transport (IFT) in living vertebrates has posed challenges, limiting our knowledge in this area.

In this study, we capitalized on the benefits of embryonic transparency in zebrafish to explore dynamic IFT in various types of cilia. To the best of our knowledge, this is the first report of IFT investigation in multiple organs within a living organism. Our findings revealed a positive correlation between the speed of IFT transport and cilia length. Interestingly, in the cilia of crista and spinal canal, ultra-high-resolution microscopy suggested an association between ciliary length and the size of IFT fluorescent particles. Our data imply the presence of a novel regulatory mechanism governing IFT speed and ciliary length control in vertebrate cilia.

## Results

### Zebrafish provide an ideal model to compare ciliogenesis in different organs

Similar to humans, cilia are widely present in various organs in zebrafish. Utilizing an Arl13b-GFP transgene under the control of the beta-actin promoter, we can observe cilia in live embryos within organs such as Kupffer's vesicle, ear, lateral line, spinal cord, and skin. Cilia in the pronephric duct, olfactory pit, photoreceptor cells, and sperms can be visualized through antibody staining with acetylated-tubulin. Notably, the number and length of cilia exhibit significant variation across different tissues

(*Figure 1*). While most cells contain a single cilium, certain specialized cells, including those in olfactory epithelium and pronephric duct, form multicilia. In adult zebrafish, multiciliated cells can also be found in the brain ventricles (ependymal cells) and ovary (*Liu et al., 2023*; *Ogino et al., 2016*). Consequently, zebrafish provides an ideal model for investigating ciliogenesis in diverse cell types (*Leventea et al., 2016*; *Song et al., 2016*).

## Rescue of *ovl* (*ift88*) mutants with *Tg(hsp70l: ift88-GFP)* transgene

To visualize IFT in zebrafish, we first generated a stable transgenic line, *Tg(hsp70l: ift88-GFP)*, which expresses a fusion protein of Ift88 and GFP under the control of the heat shock promoter (*Figure 2A*). Ift88 plays a crucial role as a component of the IFT complex in maintaining ciliogenesis. Zebrafish *ovl* (**ift88**) mutants exhibited body curvature defects and typically do not survive beyond 7 days post-fertilization (dpf) (*Tsujikawa and Malicki, 2004*). By performing a daily heat shock starting from 48 hours post-fertilization (hpf), we observed that the body curvature defects were completely rescued at 5dpf (*Figure 2B and C*). Furthermore, we assessed ciliogenesis defects in these mutants. At 5 dpf, cilia were absent in most tissues of the *ovl* mutants (*Figure 2D*). In contrast, this transgene effectively rescued ciliogenesis defects in all examined tissues (*Figure 2D*, *Figure 2—figure supplement 1*). Interestingly, the GFP fluorescence of the transgene was prominently enriched in the cilia (*Figure 2D*, *Figure 2—figure supplement 1*). Additionally, we conducted continuous heat shock experiments, which showed that *ovl* mutants carrying this transgene were able to survive to adulthood (*Figure 2—figure supplement 1B*). Taken together, these findings demonstrate that Ift88-GFP can substitute for endogenous Ift88 in promoting ciliogenesis.

## Visualization of IFT in different cilia

The enrichment of Ift88-GFP within cilia implies that the *Tg(hsp70l: ift88-GFP)* transgene could serve as a valuable tool for real-time observation of IFT movement (*Figure 2D*). For this purpose, wild type zebrafish embryos containing *Tg(hsp70l: ift88-GFP)* transgene were heat shocked at 37 °C for 4 hr to induce the expression of Ift88-GFP either at 24 hpf or 4 dpf (*Figure 3—figure supplement 1A*). Despite cilia typically being situated in the deep regions of the body, we successfully detected the movement of GFP fluorescence particles using a high-sensitive spinning-disc microscope. First, we focused on ear cristae hair cell cilia which are longer and easy to detect. We succeeded in the direct visualization of the movement of fluorescent particles in these cilia (*Figure 3—video 1*). Kymograph analysis revealed bidirectional movement of the Ift88-GFP particles along the cilia axoneme (*Figure 3A*), akin to IFT movements observed in other species (*Kozminski et al., 1993*; *Orozco et al., 1999*). Moreover, we also captured the dynamic movement of these Ift88-GFP particles in various cilia, including those of neuromasts, pronephric duct, spinal cord, and skin epidermal cells (*Figure 3B–E*, *Figure 3—videos 2–5*).

## Increased IFT velocity in longer cilia

Next, we quantified the velocity and frequency of IFT using kymographs generated from recorded movies. Both manual and automatic tracking methods showed that the retrograde IFT exhibited higher speed compared to anterograde transport within each type of cilia (*Figure 3F, G*, *Figure 3—figure supplement 1B-D*), which is consistent with previous findings in *C. reinhardtii* and *C. elegans* (*Iomini et al., 2001*; *Signor et al., 1999*; *Snow et al., 2004*). Surprisingly, we observed significant variability in IFT velocities among different cilia. In ear crista cilia, the average speed of anterograde IFT was 0.68 μm/s, while in the cilia of neuromast hair cells, it decreased to 0.54 μm/s. In skin epidermal cilia and spinal canal cilia, the transport rates were further reduced to 0.42 μm/s and 0.33 μm/s, respectively. The pronephric duct cilia showed intermediate average anterograde IFT rates at 0.46 μm/s. Similarly, retrograde transport also displayed considerable variability, with ear crista and neuromast cilia exhibiting the highest speeds (1.55 μm/s and 1.47 μm/s, respectively). In contrast, cilia in the spinal canal showed the slowest IFT, with an average speed of 0.35 μm/s, which was less than one-fourth of the speed observed in ear crista cilia (*Figure 3G*).

Notably, both anterograde and retrograde IFT displayed remarkably high transport speeds in ear crista and neuromast cilia. We observed that these cilia were particularly longer compared to others. To investigate this correlation further, we compared ciliary length in different tissues and discovered a strong correlation between cilia length and IFT speeds (*Figure 3H, I*). Specifically, longer

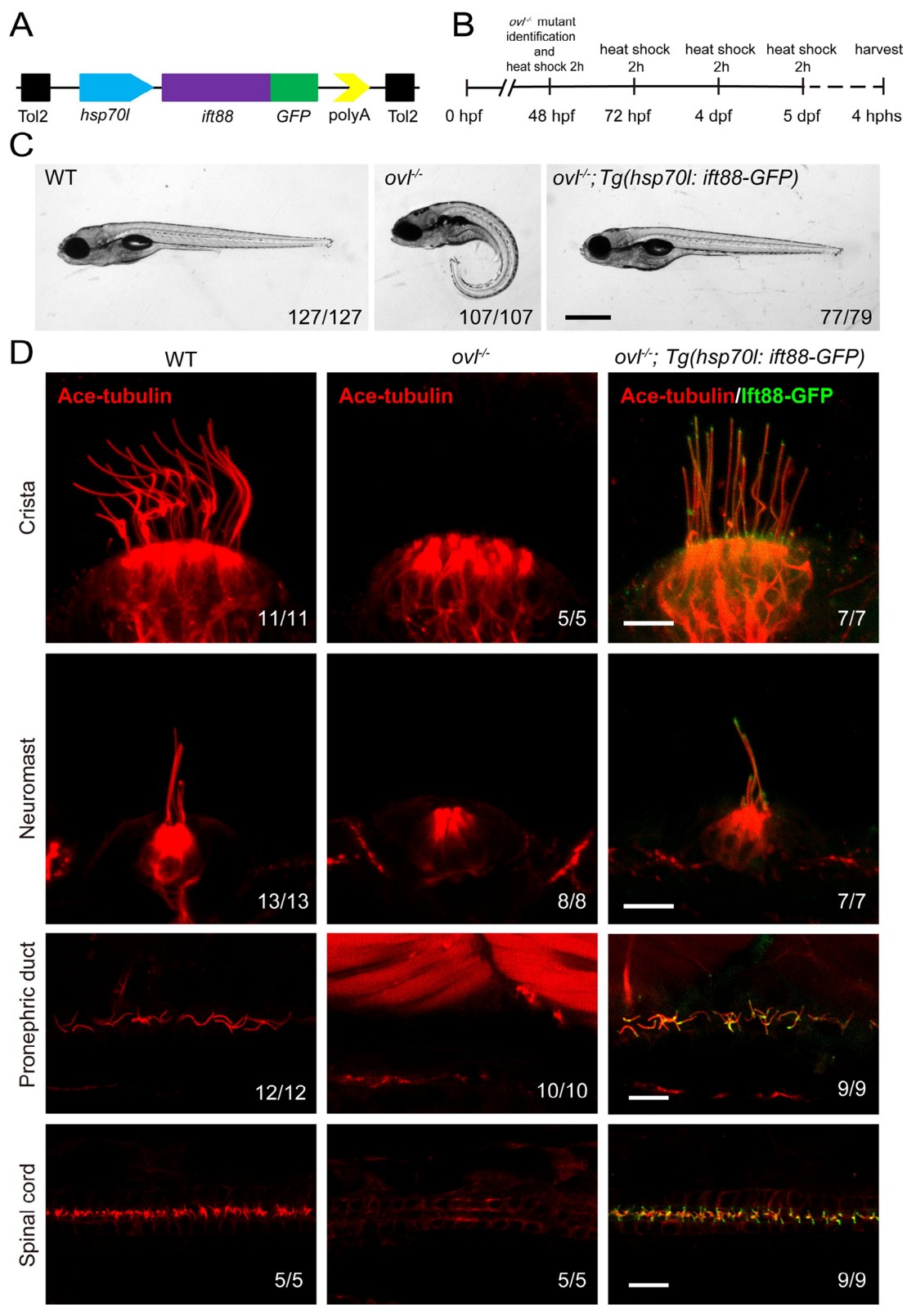

**Figure 2.** Rescue of *ovl* mutants with Tg(*hsp70l:ift88-GFP*) transgene. (**A**) Schematic diagram of *hsp70l:ift88-GFP* construct. (**B**) Procedure of heat shock experiments for *ovl* mutants rescue assay. hphs, hour post heat shock. (**C**) External pheontype of 5 dpf wild type, *ovl* mutant or *ovl* mutant larvae carrying Tg(*hsp70l:ift88-GFP*) transgene. The numbers of larvae investigated were shown on the bottom right. (**D**) Confocal images showing cilia in different type

*Figure 2 continued on next page*

*Figure 2 continued*

of organs as indicated. Red channel indicates cilia visualized by anti-acetylated α-tubulin antibody and fluorescence of *Tg(hsp70l:ift88-GFP)* is showed in green. Scale bars: 200 µm in panel C and 10 µm in panel D.

The online version of this article includes the following figure supplement(s) for figure 2:

**Figure supplement 1.** Rescue of ciliogenesis defects in *ovl* (*ift88*) mutants via *Tg(hsp70l:ift88-GFP)*.

cilia demonstrated faster IFT rates in both the anterograde and retrograde directions. Interestingly, longer cilia also have a higher frequency of fluorescent particles entering cilia (*Figure 3J*). To validate this relationship further, we created an additional transgenic line, *Tg(βactin: tdTomato-ift43)*, which allowed us to label Ift43, a constituent of the IFT-A complex. Through the analysis of Ift43 transport in this transgenic line, we reaffirmed that the tdTomato-Ift43 fluorescence particles also exhibited the highest transport speeds in ear crista cilia (*Figure 3—figure supplement 2*).

In *C. elegans*, retrograde transport has been shown to exhibit a triphasic movement pattern (*Yi et al., 2017*). We further plotted IFT particle's velocity along the entire length of crista cilia, which reveals that anterograde IFT maintains a relatively constant speed, whereas retrograde IFT undergoes a slight acceleration process when returning to the base (*Figure 3—figure supplement 1E*). Next, we compared IFT velocities in cilia of varying lengths within the same ciliary type. In the crista cilia of 4 dpf zebrafish larvae, both longer and shorter cilia were observed (*Figure 3—figure supplement 3*). Interestingly, IFT velocities were similar between the longer and shorter cilia (*Figure 3—figure supplement 3*). We also compared IFT movement in the same type of cilia at different developmental stages. In larvae at 30 hpf or 4 dpf, the length of cilia in epidermal cells is comparable. Similarly, single cilia in the pronephric duct were similar between 30 hpf and 48 hpf zebrafish larvae. In both cases, IFT speed was comparable between the two different stages (*Figure 3—figure supplement 4*). Together, these findings suggest that IFT speeds are more related to cell types in zebrafish, rather than the growth of cilia or developmental stages.

## IFT speed remains unchanged in the absence of Kif17, Kif3b, or Bbsome proteins

To understand the potential mechanisms underlying the variation in IFT speeds among different cilia, we initially focused on differences in motor proteins. In *C. elegans*, the homodimeric kinesin-2, OSM-3, drives the IFT complex at a relatively higher speed compared to the heterotrimeric Kinesin-2 (*Ou et al., 2005*). We examined IFT in *kif17* mutants, which carry a mutation of the fast homodimeric kinesin (*Zhao et al., 2012*). In the absence of Kif17, the IFT speeds remained similar to those of control larvae (*Figure 4A, B*, *Figure 4—figure supplement 1*, *Figure 4—video 1*, *Supplementary file 1*). Similarly, the IFT maintained regular speed in the absence of Kif3b in the ear crista (*Figure 4A and B*, *Figure 4—video 2*, Appendix Figure S1), possibly due to the redundant function of Kif3c in the heterotrimeric Kinesin-2 (*Zhao et al., 2012*). Additionally, we assessed IFT in the *bbs4* mutants, which has been proposed to affect IFT movements both in nematodes and mice (*Uytingco et al., 2019*). Once again, the Bbs4 mutation had little effect on IFT motility (*Figure 4A, B*, *Figure 4—figure supplement 1*, *Figure 4—video 3*, *Supplementary file 1*).

## Tubulin modifications, ATP concentration and IFT

The post-translational modifications of axonemal tubulins can affect the interaction between microtubules and motor proteins, thereby regulating their dynamics (*Hong et al., 2018*; *Janke and Bulinski, 2011*; *O'Hagan et al., 2017*, *Sirajuddin et al., 2014*). Next, we asked whether tubulin modifications can affect IFT in zebrafish. Ttll3 is a tubulin glycylase that are involved in the glycylation modification of ciliary tubulin (*Wloga et al., 2009*). We generated zebrafish *ttll3* mutants and identified a mutant allele with an 8 bp insertion, which causes frameshift, resulting in a significantly truncated Ttll3 protein without TTL domain (*Figure 4C and D*). Immunostaining with anti-monoglycylated or polyglycylated tubulin antibody revealed that the glycylation modification was completely eliminated in all the cilia investigated (*Figure 4E*, *Figure 4—figure supplement 2A, B*). Surprisingly, there are no obviously defects in the number and length of cilia in the mutants. The velocity of IFT within different cilia also appears unchanged (*Figure 4F*, *Figure 4—video 4*, Appendix Figure S1). The beating pattern of cilia in the pronephric is similar between control and

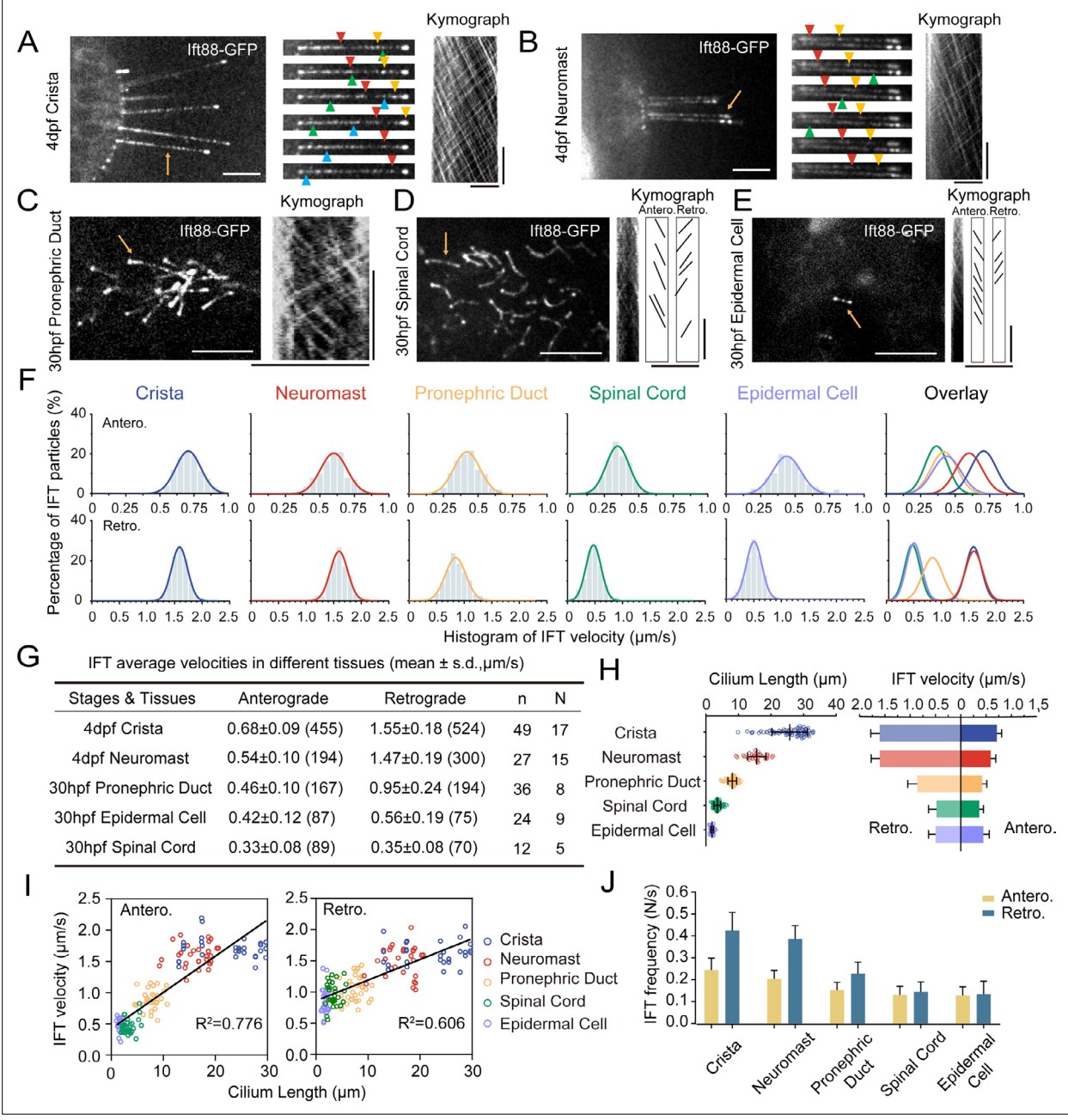

**Figure 3.** Intraflagellar transport in different type of cilia. (**A–E**) Left, Snapshot of Intraflagellar transport videos in different cell types as indicated. Middle (**A, B**) Snapshot of same cilia at different time points. Arrowheads with the same color indicate the same IFT particle. Right, kymographs illustrating the movement of IFT particles along the axoneme. Horizontal scale bar: 10 μm and vertical scale bars: 10 s. Representative particle traces are marked with black lines in panels D and E. Yellow arrows denote the cilia used to generating the kymograph. (**F**) Histograms displaying the velocity of anterograde and retrograde IFT in different type of cilia as indicated. 'Antero.' and 'Retro.' represent anterograde and retrograde transport, respectively. Each plot was fit by a Gaussian distribution. The developmental stages of zebrafish were consistent with (**A–E**) (**G**) Summary of IFT velocities in different tissues of zebrafish. Numbers of IFT particles are shown in the brackets. n, number of cilia detected. N, number of zerafish larvae analyzed. hpf, hours post-fertilization. dpf, days post-fertilization. (**H**) Left, Statistics analysis of cilia length in different tissues of Tg(*hsp70l:ift88-GFP*) larvae (crista, n=72 cilia from 16 larvae; neuromast, n=31 cilia from 9 larvae; pronephric duct, n=72 cilia from 11 larvae; spinal cord, n=86 cilia from 6 larvae; epidermal cell, n=48 cilia from 18 larvae). Average cilium length: crista, 25.46 μm; neuromast, 15.2 μm; pronephric duct, 8.1 μm; spinal cord, 1.03 μm; epidermal cell, 0.49 μm. Right, anterograde and retrograde IFT average velocity in different tissues of zebrafish. (**I**) Anterograde and retrograde IFT velocities plotted versus cilia length. Linear fit (black line) and coefficient of determination are indicated. (**J**) Frequency of anterograde and retrograde IFT entering or exiting cilia. (crista, n=47 cilia from 13 larvae; neuromast, n=31 cilia from 13 larvae; pronephric duct, n=43 cilia from 14 larvae; spinal cord, n=27 cilia from 7 larvae; epidermal cell, n=26 cilia from 15 larvae).

*Figure 3 continued on next page*

*Figure 3 continued*

The online version of this article includes the following video, source data, and figure supplement(s) for figure 3:

**Source data 1.** Raw data used to generate *Figure 3F, H–J*.

**Figure supplement 1.** Overview of zebrafish larvae treatments and IFT velocity analysis.

**Figure supplement 1—source data 1.** Raw data used to generate *Figure 3—figure supplement 1C, E*.

**Figure supplement 2.** Generation of Tg (*βactin:tdTomato-ift43*) transgene for IFT imaging.

**Figure supplement 2—source data 1.** Raw data used to generate *Figure 3—figure supplement 2D*.

**Figure supplement 3.** IFT velocity in cilia of different lengths within crista.

**Figure supplement 3—source data 1.** Raw data used to generate *Figure 3—figure supplement 3*.

**Figure supplement 4.** Comparison of cilia length and IFT velocity at different developmental stages.

**Figure supplement 4—source data 1.** Raw data used to generate *Figure 3—figure supplement 4C, D, H, I*.

**Figure 3—video 1.** Fluorescence time-lapse movie of ear crista cilia visualized by *Tg(hsp70l: ift88-GFP)*.
https://elifesciences.org/articles/93168/figures#fig3video1

**Figure 3—video 2.** Fluorescence time-lapse movie of neuromast cilia visualized by *Tg(hsp70l: ift88-GFP)*.
https://elifesciences.org/articles/93168/figures#fig3video2

**Figure 3—video 3.** Fluorescence time-lapse movie of pronephric duct cilia visualized by *Tg(hsp70l: ift88-GFP)*.
https://elifesciences.org/articles/93168/figures#fig3video3

**Figure 3—video 4.** Fluorescence time-lapse movie of spinal cord cilia visualized by *Tg(hsp70l: ift88-GFP)*.
https://elifesciences.org/articles/93168/figures#fig3video4

**Figure 3—video 5.** Fluorescence time-lapse movie of skin epidermal cells cilia visualized by *Tg(hsp70l: ift88-GFP)*.
https://elifesciences.org/articles/93168/figures#fig3video5

mutant larvae (*Figure 4—figure supplement 2C–E*). Moreover, the mutants were able to survive to adulthood and there is no difference in the fertility or sperm motility between mutants and control siblings, which is slightly different to those observed in mouse mutants (*Figure 4—figure supplement 2F–H*; *Gadadhar et al., 2021*).

Next, we investigated whether polyglutamylation of axonemal tubulins can regulate IFT movement in zebrafish. First, we knocked down the expression of *ttll6*, which has been shown earlier to reduce tubulin glutamylation and resulted in body curvature in zebrafish (*Figure 4—figure supplement 3A, B*; *Pathak et al., 2011*). The efficiency of *ttll6* morpholinos was confirmed via RT-PCR (reverse transcription-PCR), which showed the splicing error of intron 12 when introduced the morpholinos (*Figure 4—figure supplement 3C, D*). Interestingly, we observed only minor differences in IFT velocity between *ttll6* morphants and control groups (*Figure 4G*, Appendix Figure S1). Ccp5 is the major tubulin deglutamylase that regulates glutamylation levels inside cilia (*Pathak et al., 2014*). Interestingly, the IFT speeds exhibited only slight changes in *ccp5* morphants (*Figure 4G*, *Figure 4—figure supplement 3E-H*, *Supplementary file 1*). Together, these results suggest that glycylation and glutamylation modification may play relatively minor roles on IFT in zebrafish. Considering the big difference in the IFT speed among different cilia, we think it is very unlikely such difference is caused by different levels of tubulin glycylation or glutamylation modification.

The movement of kinesin and dynein motors relies on the energy derived from ATP hydrolysis. *In vitro* studies using molecular force clamp techniques have shown that increasing ATP concentration significantly enhances the speed of kinesin during cargo loading (*Visscher et al., 1999*). We further examined whether the difference of IFT speeds was caused by variation of ATP concentration inside cilia. We generated an ATP reporter line, *Tg (βactin: arl13b-mRuby-iATPSnFR$^{1.0}$)*, which contains a ciliary localized ATP sensor, mRuby-iATPSnFR$^{1.0}$ driven by β-actin promotor (*Figure 4—figure supplement 4A–B*; *Lobas et al., 2019*). This ATP sensor utilizes relative fluorescence intensity to indicate the concentration of ATP. By measuring the ratio of mRuby red fluorescence to green fluorescence in the longer crista and shorter epidermal cilia, we compared relative ATP levels between these two types of cilia. Again, we found no difference in the ATP concentration between crista and epidermal cilia, suggesting that variation of ATP concentration was unlikely to be the cause of different IFT speed (*Figure 4—figure supplement 4C*).

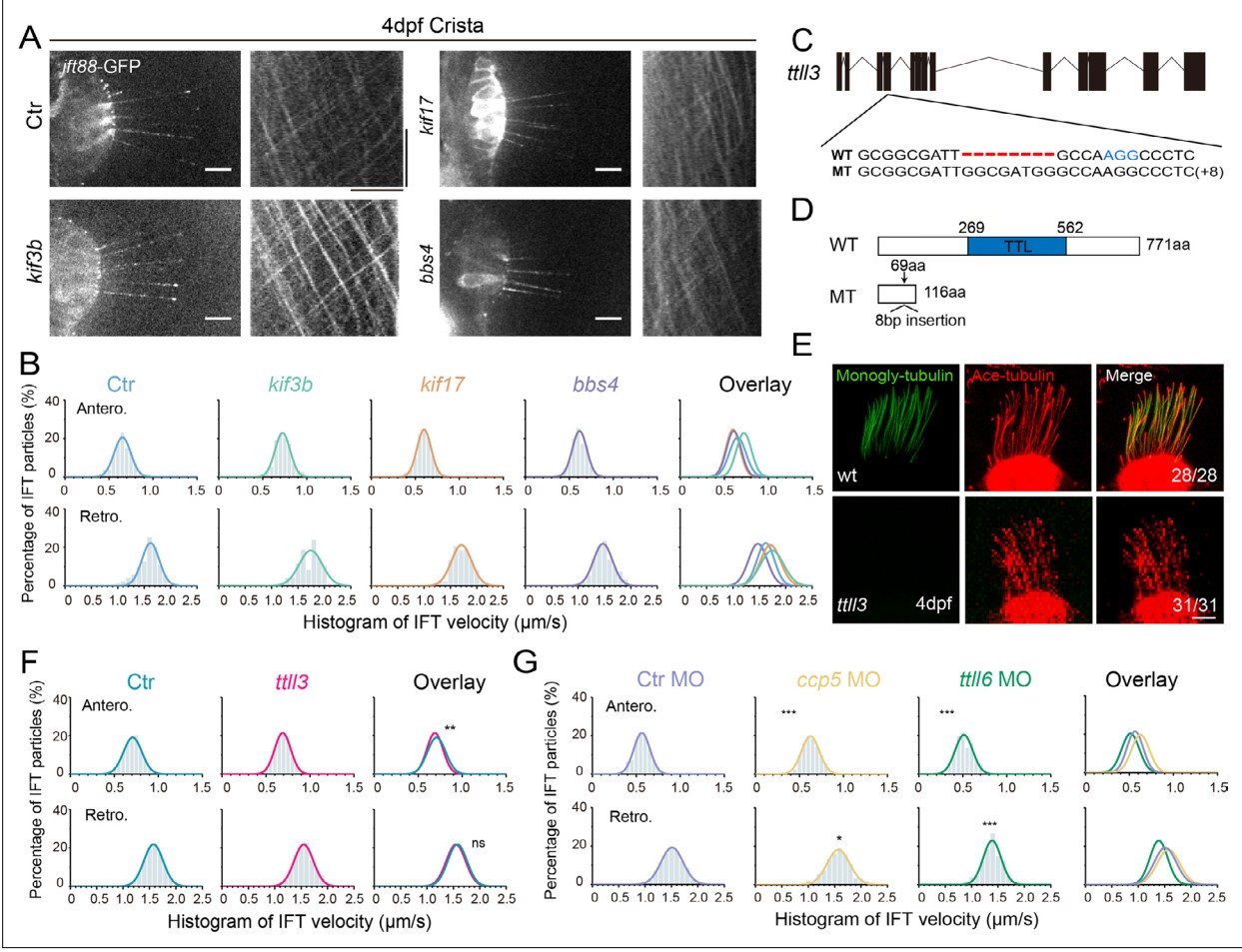

**Figure 4.** Alterations in motor proteins, BBSome proteins, or tubulin modifications have minimal effects on IFT. (**A**) Left: Snapshot of IFT videos in crista cilia of 4dpf wild type or mutant larvae as indicated. Right: Kymographs showing IFT particle movement along axoneme visualized with Ift88:GFP. Horizontal scale bar: 10 μm and vertical scale bar:10 s. (**B**) Histograms showing anterograde and retrograde IFT velocity in crista cilia of control or mutant larvae. (**C**) Genomic structure and sequences of wild type and *ttll3* mutant allele. PAM sequence of sgRNA target are indicated in blue. (**D**) Protein domain of Ttll3 in wild type and *ttll3* mutants. (**E**) Confocal images showing crista cilia in 4dpf wild type (wt) or maternal-zygotic (MZ) *ttll3* mutant visualized with anti-monoglycylated tubulin antibody (green) and anti-acetylated α-tubulin antibody (red). (**F**) Histograms depicting IFT velocity in crista cilia of control and *ttll3* mutants. Top, anterograde IFT. Bottom, retrograde IFT.(**G**) Histograms illustrating IFT velocity in crista cilia of *ccp5* or *ttll6* morphants. Scale bars: 10 μm in panel A and 5 μm in panel E. *$p < 0.5$; *** $p < 0.001$. The average IFT velocity and the number of samples (**N**) are shown in *Supplementary file 1*.

The online version of this article includes the following video, source data, and figure supplement(s) for figure 4:

**Source data 1.** Raw data used to generate *Figure 4B, F, G*.

**Figure supplement 1.** IFT in the cilia of neuromast hair cells of different zebrafish mutants.

**Figure supplement 1—source data 1.** Raw data used to generate *Figure 4—figure supplement 1B*.

**Figure supplement 2.** Loss of tubulin glycylation in *ttll3* mutants.

**Figure supplement 2—source data 1.** Raw data used to generate *Figure 4—figure supplement 2E, F*.

**Figure supplement 3.** Validation of the efficiency of *ttll6* and *ccp5* morpholinos.

**Figure supplement 3—source data 1.** PDF file containing original gels for *Figure 4—figure supplement 3*, indicating the relevant bands.

**Figure supplement 3—source data 2.** Original files for gels displayed in *Figure 4—figure supplement 3C, G*.

**Figure supplement 4.** Generation of ATP reporter transgenic line.

**Figure supplement 4—source data 1.** Raw data used to generate *Figure 4—figure supplement 4C*.

**Figure 4—video 1.** Fluorescence time-lapse movies of ear crista cilia in *kif17*, *kif3b*, *bbs4*, and *ttll3* mutants visualized by *Tg(hsp70l: ift88-GFP)*.
https://elifesciences.org/articles/93168/figures#fig4video1

**Figure 4—video 2.** Fluorescence time-lapse movies of ear crista cilia in *kif17*, *kif3b*, *bbs4*, and *ttll3* mutants visualized by *Tg(hsp70l: ift88-GFP)*.

*Figure 4 continued on next page*

*Figure 4 continued*

https://elifesciences.org/articles/93168/figures#fig4video2

**Figure 4—video 3.** Fluorescence time-lapse movies of ear crista cilia in *kif17*, *kif3b*, *bbs4*, and *ttll3* mutants visualized by *Tg(hsp70l: ift88-GFP)*.

https://elifesciences.org/articles/93168/figures#fig4video3

**Figure 4—video 4.** Fluorescence time-lapse movies of ear crista cilia in *kif17*, *kif3b*, *bbs4*, and *ttll3* mutants visualized by *Tg(hsp70l: ift88-GFP)*.

https://elifesciences.org/articles/93168/figures#fig4video4

## Larger IFT particles are present in longer cilia

Finally, we aim to investigate the size of IFT particles within different type of cilia. In the flagellar of *Chlamydomonas*, IFT particles are usually transported as IFT trains, consisting of multiple IFT-A and IFT-B repeating subcomplexes and the kinesin-2 and IFT dynein motors. However, with current technique, it is unfeasible to distinguish the size of IFT trains on different type of cilia through ultra-structure electron microscopy in zebrafish. Instead, we sought to compare the size of the fluorescence particles using STED ultra-high-resolution microscopy. With this method, we were able to identify single IFT fluorescence particles with relatively high resolution. Compared with regular spinning disk data, the number of IFT fluorescence particles was significantly increased in the cilia (*Figure 5A and B*). When comparing the size of these fluorescence particles, we found that the particle sizes were significantly larger in the longer crista cilia than those of the shorter spinal cord cilia (*Figure 5C*).

To further test whether the higher IFT velocity was associated with large IFT particles in the crista cilia, we performed morpholino based knockdown analysis. Injection of *ift88* morpholino caused body curvature due to ciliogenesis defects (*Figure 5D*). When injecting lower dose of *ift88* morpholino, we found zebrafish embryos can still maintain grossly normal body axis, while cilia in ear crista became significantly shorter (*Figure 5D, E, I*). Due to partial loss of Ift88 proteins, the number of IFT complex decreased significantly as suggested by the reduced number of IFT fluorescence particles in the morphant cilia (*Figure 5F and G*). Strikingly, the size of the fluorescent particles also decreased significantly (*Figure 5F and H*). Moreover, the average intensity and size of fluorescent particles showed a strong correlation in both control and *ift88* morphants, which implies a potentially reduced length of the IFT trains (*Figure 5I*). Interestingly, we found the IFT speed also decreased significantly in the shorter cilia (1.55 µm/s in control vs 1.18 µm/s in morphants *Figure 5J and K*, *Figure 5—video 1*). Taken together, these data imply that the size of IFT fluorescence particles is related to IFT speed and longer cilia are prone to contain larger IFT particles than shorter cilia.

## Discussion

With its diverse array of cilia types, zebrafish provide an exceptional model for investigating the intricate mechanisms underlying ciliary length regulation. By creating a transgenic line that expresses ciliary-targeted IFT proteins, we have demonstrated a positive correlation between IFT speed and the length of cilia. To the best of our knowledge, this study represents the first instance of comparing IFT in cilia from different types of organs within the same organism. Interestingly, the relationship between ciliary length and IFT speed has been studied recently in *Chlamydomonas,* which suggested that reduction of the IFT speed with chimeric CrKinesin-II can affect ciliary assembly activity, together with the formation of slightly shorter flagella (*Li et al., 2020*).

Surprisingly, certain conventional factors known to govern IFT transport regulation in *C. elegans* or *Chlamydomonas* appear to exert limited influence in zebrafish. IFT transport speed showed no discernible connection to ciliary motility, as evidenced by the comparable IFT speeds observed in both motile spinal cord cilia and primary cilia of the skin's epidermal cells (*Figure 3*). Furthermore, IFT remains largely unchanged in mutants of *kif3b*, *kif17*, *bbs4*, or *ttll3*. In *kif3b* mutants, Kif3c may substitute for Kif3b to ensure proper IFT (*Zhao et al., 2012*). Although both KIF17 (OSM-3) and BBSome proteins have been shown to mediate IFT transport in the cilia of olfactory neurons in both nematodes and mice (*Ou et al., 2005*; *Uytingco et al., 2019*; *Williams et al., 2014*), it is still questionable whether these proteins can affect IFT in other types of cilia. Our data suggest that IFT velocity was not altered in the crista or neuromast cilia of *kif17* or *bbs4* mutants, consistent with the fact that Kif17 and BBSome proteins are dispensable for ciliogenesis in these tissues. Unfortunately, we couldn't detect IFT trafficking in the olfactory neurons of zebrafish with current techniques. It would be worthwhile to

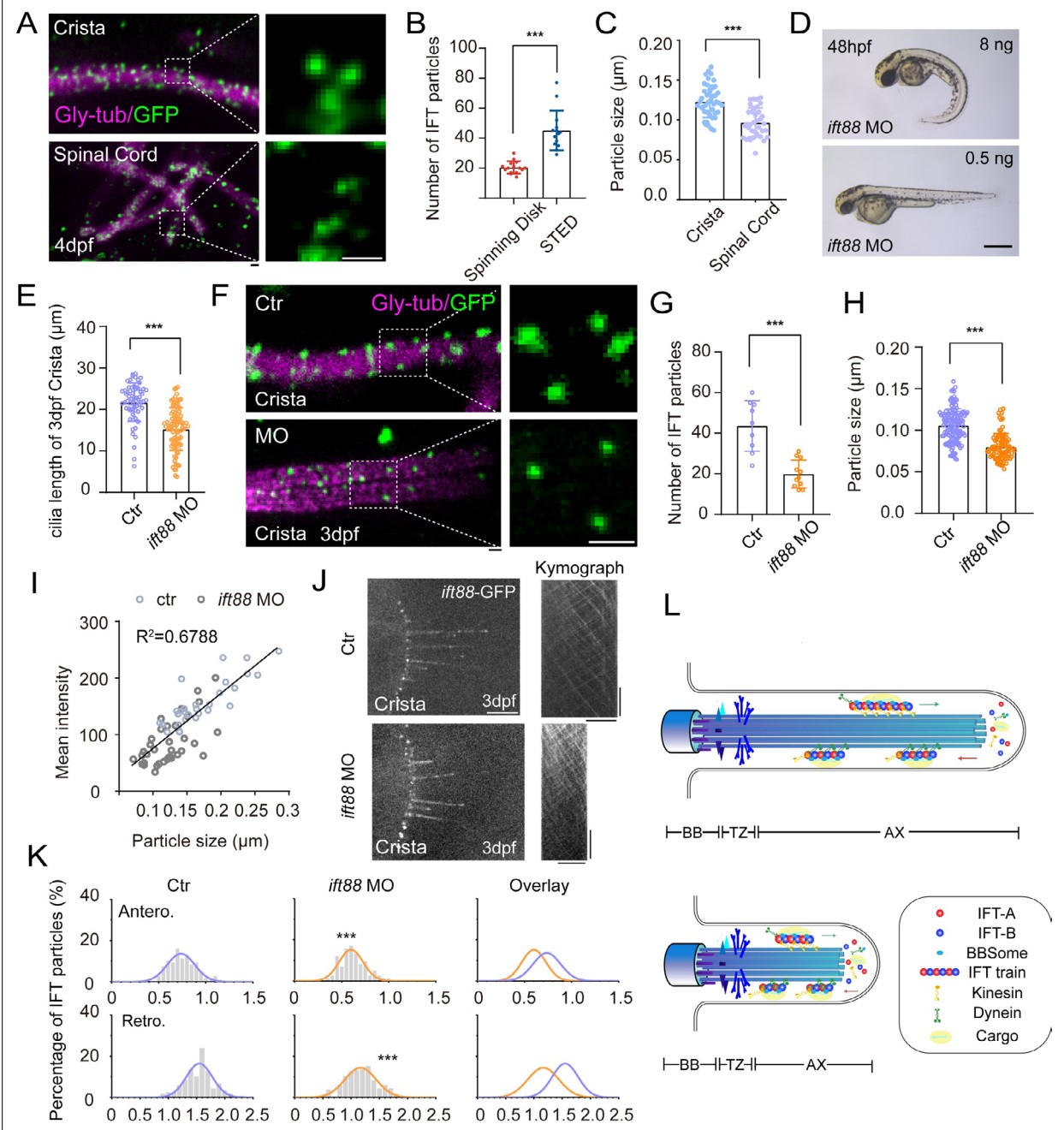

**Figure 5.** Increased size of IFT fluorescent particles in crista cilia. (**A**) Representative STED images of crista (top) and spinal cord (bottom) in 4dpf *Tg(hsp70l: ift88-GFP)* larva. Cilia was stained with anti-monoglycylated tubulin (magenta), and IFT88-GFP particles were counterstained with anti-GFP antibody (green). Enlarged views of the boxed region are displayed on the right. (**B**) Dot plots showing the number of IFT particles per cilia in crista recorded by spinning disk and STED. (Spinning disk, n=15 cilia from 6 larvae; STED, n=14 cilia from 8 larvae.Two-tailed Mann-Whitney test) (**C**) Statistical analysis showing IFT particles size in the cilia of ear crista and spinal cord. (Crista, n=44 particles from 7 larvae; Spinal cord, n=35 particles from 5 larvae; Unpaired two-sided Student's t-test) (**D**) External phenotypes of 2 dpf zebrafish larvae injected with higher and lower dose of *ift88* morpholinos. (**E**) Cilia length quantification of control and *ift88* morphants. (ctr, n=62 cilia from 18 larvae; *ift88* MO, n=88 cilia from 17 larvae; Two-tailed Mann-Whitney test) (**F**) STED images showing IFT particles in crista cilia of 3dpf control or *ift88* morphants. Enlarged views of the boxed region are displayed on the right. (**G**) Dot plots showing the number of IFT particles per cilia in control and *ift88* morphants. (ctr, n=9 cilia from 5 larvae; *ift88* MO, n=11 cilia from 6 larvae; Unpaired two-sided Student's t-test) (**H**) Statistical analysis showing IFT particles size of crista cilia in control or *ift88* morphants. (ctr, n=141 particles from 11 larvae; *ift88* MO, n=88 particles from 13 larvae; Two-tailed Mann-Whitney test) (**I**) Mean intensity of IFT particles was plotted together with IFT particle size. Linear fit (black line) and coefficient of determination are indicated. (**J**) Left, Snapshot of IFT videos in crista cilia of 3dpf control (top) or *ift88* morphant (bottom) carrying *Tg(hsp70l: ift88-GFP)*. Right, Kymographs showing movement of IFT particles along axoneme. Horizontal scale bar:

*Figure 5 continued on next page*

*Figure 5 continued*

10 μm and vertical scale bar:10 s. (**K**) Histograms showing IFT velocity in the crista cilia of 3dpf control or *ift88* morphants. (**L**) Model illustrating IFT with different train sizes in long and short cilia. Scale bars: 0.2 μm in panel A and F, and 500 μm in panel I. \*\*p<0.01; \*\*\* p<0.001. BB, basal body; TZ, transition zone; AX, axoneme.

The online version of this article includes the following video and source data for figure 5:

**Source data 1.** Raw data used to generate *Figure 5B, C, E, G–I and K*.

**Figure 5—video 1.** Fluorescence time-lapse movies of ear crista cilia in *Tg(hsp70l: ift88-GFP)* embryos injected with lower dose of *ift88* morpholinos.
https://elifesciences.org/articles/93168/figures#fig5video1

improve detection methods to determine whether the role of Kif17 or BBSome proteins is conserved during the ciliogenesis of olfactory neurons. Finally, although tubulin modifications are essential for IFT in *C. elegans*, IFT remains unchanged in *ttll3* mutants, in which tubulin glycylation is completely removed. Similarly, modification of tubulin glutamylation levels has minor effects on the movement of IFT. Collectively, these results suggest that the regulation of IFT velocity in zebrafish cilia differs from that observed in *C. elegans* and mice OSNs, thereby highlighting the intricate complexity of IFT regulation across various organisms.

Remarkably, our studies revealed that the IFT fluorescence particles were significantly larger in longer crista cilia than those of shorter cilia. The IFT complex typically travels as trains, with multiple repeating IFT units being transported simultaneously. The increased size of the fluorescence particles implies that longer cilia might form larger IFT trains for more effective cargo transportation. Decreasing the quantity of IFT88 proteins could diminish the likelihood of IFT complex assembly, consequently leading to a reduction in the size of IFT trains. Importantly, the size of IFT particles was significantly reduced in *ift88* morphants, concurrent with the decreased transport speed of IFT. Thus, our data support a length control model for cilia that operates through the modulation of IFT train size (*Figure 5L*). Within longer cilia, the IFT complex appears predisposed to form lengthier repeating units, thereby creating an optimal platform for efficient transport. This environment enhances the coordination between cargos and motor proteins, resulting in an improved transportation speed. This orchestration potentially involves the synchronization of motor proteins, ensuring their precise functionality and directional alignment during transport—a concept supported by various models (*Stukalin et al., 2005*; *Urnavicius et al., 2018*). Furthermore, insights from the cryo-EM structure of intraflagellar transport trains have suggested that each dynein motor might propel multiple IFT complexes. For instance, the ratio of dynein: IFT-B: IFT-A in the flagella of *Chlamydomonas* is approximately 2:8:4 (*Jordan et al., 2018*). It remains plausible that longer cilia in vertebrates could recruit a higher ratio of motor proteins to execute IFT, thus intensifying the driving force for cargo transport.

Finally, we want to mention that this study has several potential limitations. For instance, the evaluation of IFT particle sizes was based on immunostaining results acquired from STED microscopy, and we cannot distinguish between anterograde or retrograde IFT transport as can be done with live imaging. It remains unclear whether IFT particles are larger in both directions. Moreover, the size difference of IFT particles was only compared between crista and spinal cord cilia. Due to technical issues, many cilia, such as those in epidermal cells, were difficult to identify under high magnifications. It would be helpful to isolate these cilia and perform ultrastructural analysis to compare IFT train sizes between different types of cilia. Additionally, although IFT speed was largely normal in the absence of Kif3b, Bbs4, and tubulin glycylation modification, it remains possible that other factors, such as the presence of different adaptor proteins in different cell types or the use of different tubulin codes, are involved in mediating IFT. The mechanisms behind these conditions may require further investigation.

In summary, due to their relatively simple structure and ease of measurement, cilia provide an excellent model for investigating the mechanisms of organelle size control. Our results indicate that cilia in zebrafish vary in length, and this length is closely related to cell types. Longer cilia may form larger IFT trains for faster transport to maintain their stability. It is possible that a larger IFT pool is present in the base region of longer cilia, which help assemble large IFT trains. The increased frequency of IFT transport in longer cilia further supports this concept. Comparing the expression levels of IFT and motor proteins across different cell types will be helpful in elucidating the differences in IFT injection rates among these cilia. Interestingly, the relationship between IFT transport velocity and cell type is also observed in other cell types, such as mouse olfactory neurons. In these neurons, cilia of the

same type maintain similar IFT velocities, while different types of olfactory neurons exhibit significant variations (*Williams et al., 2014*). Investigating the IFT velocity within different cilia in mouse models in the future will be valuable.

## Materials and methods

### Zebrafish strains

Zebrafish Tuebingen (TU) strains were maintained at 28 °C on a 14 hr/10 hr light/dark cycle. Embryos were raised at 28.5 °C in E3 medium (5 mM NaCl, 0.17 mM KCl, 0.39 mM CaCl2, 0.67 mM MgSO4). The following mutant strains were used: *ovl* (*Tsujikawa and Malicki, 2004*), *kif3b* and *kif17* (*Zhao et al., 2012*). Three transgenic lines, *Tg(hsp70l:ift88-GFP), Tg(βactin:tdTomato-ift43)* and *Tg(βactin: arl13b-mRuby-iATPSnFR$^{1.0}$)*, were generated in this study. The constructs for making these transgene were created using Tol2 kit Gateway-based cloning (*Kwan et al., 2007*). The *ttll3* mutants were generated via CRISPR/Cas9 technology with the following target sequence: 5'-GGGTGGAGCGGCGATT GCCA-3'. The *bbs4* mutants were generated by TALEN system with the following binding sequences: 5'-TAAACTTGGCATTACAGC-3' and 5'- TCCCAGCATCATGTAGGTC –3'. All strains involved for analysis are stable lines.

### IFT imaging

To prepare for time-lapse imaging, *Tg (hsp70l: ift88-GFP*) embryos were subjected to single heat shock treatment for 4 hr at 37 °C at either 24 hpf or 4dpf (*Figure 3—figure supplement 1A*). Then zebrafish embryos were raised at 28.5 °C in fresh E3 medium for another 2–4 hr to obtain brighter fluorescence and higher signal to noise ratio. For living imaging, we used 35 mm glass-bottom dishes (FD35-100, WPI) filled with a thin layer of 3% agarose. Once the agarose solidified, several rectangle slots were cut out from dish using a capillary pipette. Zebrafish larvae were anesthetized with E3 water containing 0.01% Tricaine, and put into slots. After removing excess water, 1% low melting point agarose was added to cover the embryos. Finally, E3 water containing 0.01% Tricaine was added to the dish. IFT movement in cilia of crista and neuromast were recorded using an Olympus IX83 microscope equipped with a 60 X, 1.3 NA objective lens, an EMCCD camera (iXon +DU-897D-C00-#BV-500; Andor Technology) and a spinning disk confocal scan head (CSU-X1 Spinning Disk Unit; Yokogawa Electric Corporation). Time-lapse images were continuously collected with 100 or 200 repeats using μManager (https://www.micro-manager.org) at an exposure time of 200ms. IFT movement in cilia of epidermal cells, spinal cord, and pronephric duct epithelial cells were recorded using an Olympus IX83 microscope equipped with a 100 X, 1.49 NA objective lens, and the same EMCCD camera and spinning disk confocal modules as mentioned above. We usually collected IFT images within 5 min after embedding and only single focal plane images were used for analysis.

### Image processing and analysis

ImageJ software was used to generate kymographs for quantifying cilium length and velocity. To analyze IFT velocity, we employed an established method as previously described (*Zhou et al., 2001*). We first manually drew lines along the cilia to define a track for detailed analysis of IFT velocity. Using a Fourier filtering algorithm, the KymographClear plugin automatically color-coded anterograde and retrograde particles. IFT velocities at different positions along the cilia were automatically extracted using KymographDirect software and classified into groups at 2 μm intervals along the cilia. These velocities were then quantified to generate the anterograde and retrograde velocity curves (*Mangeol et al., 2016*).

### Whole-mount Immunofluorescence

Zebrafish larvae were fixed overnight at 4 °C in 4% PFA. After removing the fixative, they were washed three times with PBS containing 0.5% Tween 20 (PBST) and then incubated in acetone for 10 min for permeabilization. Subsequently, the embryos were treated with a blocking reagent (PBD with 10% goat serum) for 1 hr and sequentially labeled with primary and secondary antibodies (at a 1:500 dilution) overnight at 4 °C. The following antibodies were used: anti-acetylated α-tubulin (Sigma T6793), anti-SYCP3 (Abcam, ab150292), anti-monoglycylated tubulin antibody, clone TAP 952 (Sigma MABS277),

anti-polyglycylated tubulin antibody, clone AXO49 (Sigma MABS276), anti-GFP (Invitrogen A-11120), Alexa Fluor 633 phalloidin (Invitrogen A22284), and zpr-1 (Zebrafish International Resource Center).

### Morpholino knockdown

The following morpholinos were used: *ttll6* (5' – GCAACTGAATGACTTACTGAG-TTTG - 3'), *ccp5* (5' -TCCTCTTAATGTGCAGATACCCGTT-3'), *ift88* (5' -CAACTCCACTCACCCCATAAGCTGT - 3'; *Tsujikawa and Malicki, 2004*) and a standard control morpholino (5' -CCTCTTACCTCAGTTACAAT TTATA-3'). All morpholinos were purchased from Gene Tools (Philomath, OR). To detect splicing defects caused by morpholinos, we extracted total RNA from 24hpf embryos and performed RT-PCR using the following primer sequences: *ttll6* Forward: 5'-AAAGTATTTCCAACACAGCAGCTC-3', Reverse: 5'-GTGGTCGTGTCTGCAGTGTGGAGG-3'; *ccp5* Forward: 5'-TCCTGTCGTTTGTTCATCGT CTGC-3', Reverse: 5'-CTTAAAGACGAACATGC- GGCGAAG-3'.

### Super-resolution microscopy

High resolution images were acquired using Abberior STEDYCON (Abberior Instruments GmbH, Göttingen, Germany) fluorescence microscope built on a motorized inverted microscope IX83 (Olympus UPlanXAPO 100 x, NA1.45, Tokyo, Japan). The microscope is equipped with pulsed STED lasers at 775 nm, and with 561 nm and 640 nm excitation pulsed lasers. Zebrafish larvae were collected and fixed in 4% PFA overnight at 4 °C, then processed for immunofluorescence with anti-GFP antibody and anti-monoglycylated tubulin antibody. For STED imaging, the following secondary antibody were used: goat anti-mouse Abberior STAR RED and goat anti-rabbit Abberior STAR ORANGE. The embryos were incubated in secondary antibody (1:100) in blocking buffer at room temperature for 2 hr. After wash three times (5 min each) with PBST, larvae were cryoprotected in 30% sucrose overnight. Sagittal sections were collected continuously at 40 µm thickness on Leica CM1850 cryostat. Place the slices at 37 °C for 1 hr to dry. Rehydrate in PBST for 5 min and cover with Abberior Mount Liquid Antifade. After sealing, the slices were placed at 4 °C overnight before imaging.

### High-speed video microscopy

Cilia beating in the zebrafish pronephric duct was recorded at 4dpf as previously described (*Xie et al., 2020*). To visualize sperm motility, adult male zebrafish were euthanized with ice-cold water. Excess water was removed from the fish's body, and the testes were carefully dissected using tweezers in cold Hank's solution (136.75 mM NaCl, 5.37 mM KCl, 0.25 mM $Na_2HPO_4$, 0.45 mM $KH_2PO_4$, 1.3 mM CaCl2, 1 mM MgSO4, 4.17 mM $NaHCO_3$). The supernatant containing the sperm was carefully collected and placed on ice. A 6 µl aliquot of the supernatant was aspirated using a pipette and mixed with 2 µl of water. This mixture was then placed on top of a cover glass. The cover glass, with the sperm sample, was inverted and positioned in the center of a depression slide. Sperm motility was recorded using a 100 X oil objective on a Leica Sp8 confocal microscope equipped with a high-speed camera (Motion-BLITZ EoSens mini1; Mikrotron, Germany), capturing images at a rate of 250 frames per second. Sperm movement trajectory was marked with manual tracking plugins in imageJ.

### Statistical analysis

Statistical analysis was performed using GraphPad Prism 8 software. Student's t-test was used when the data follow a normal distribution, while the Mann-Whitney test was used when the data do not follow a normal distribution. Data for statistical analysis are expressed as mean ±s.d. unless otherwise stated. ImageJ software was used to measure length of cilia, fluorescence intensity and particle size.

## Acknowledgements

We thank Dr. Guangshuo Ou and members of the Zhao lab for their kind help during the preparation of this manuscript. We are also grateful for the excellent support from the core facilities of IEMB at OUC. This work was supported by the National Natural Science Foundation of China (Nos. 32125015, 31991194 to C.Z. and No. 32100661 to H.X.) and the founding from Laoshan Laboratory (LSKJ202203204).

## Additional information

### Funding

| Funder | Grant reference number | Author |
|---|---|---|
| National Natural Science Foundation of China | 32125015 | Chengtian Zhao |
| National Natural Science Foundation of China | 31991194 | Chengtian Zhao |
| National Natural Science Foundation of China | 32100661 | Haibo Xie |
| China Postdoctoral Science Foundation | 2023M733344 | Haibo Xie |

The funders had no role in study design, data collection and interpretation, or the decision to submit the work for publication.

### Author contributions

Yi Sun, Data curation, Formal analysis, Validation, Investigation, Visualization, Methodology, Writing – original draft, Writing – review and editing; Zhe Chen, Data curation, Formal analysis, Validation, Investigation, Visualization, Methodology; Minjun Jin, Data curation, Formal analysis, Validation, Investigation, Visualization; Haibo Xie, Resources, Supervision, Funding acquisition, Writing – review and editing; Chengtian Zhao, Conceptualization, Resources, Supervision, Funding acquisition, Validation, Investigation, Visualization, Writing – original draft, Project administration, Writing – review and editing

### Author ORCIDs

Yi Sun ⓘ https://orcid.org/0009-0005-7295-7412
Zhe Chen ⓘ https://orcid.org/0009-0002-8988-6225
Chengtian Zhao ⓘ https://orcid.org/0000-0003-1236-914X

### Ethics

All zebrafish study was conducted according standard animal guidelines and approved by the Animal Care Committee of Ocean University of China (Animal protocol number: OUC2012316).

Reviewer #2 (Public review): https://doi.org/10.7554/eLife.93168.3.sa1
Reviewer #3 (Public review): https://doi.org/10.7554/eLife.93168.3.sa2
Author response https://doi.org/10.7554/eLife.93168.3.sa3

## Additional files

### Supplementary files

• Supplementary file 1. Summary of average IFT velocities in different zebrafish muants or morphants.
• Supplementary file 2. Detailed statistics.
• MDAR checklist

### Data availability

All data generated or analysed during this study are included in the manuscript and supporting files.

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
