## [Editor Report · eLife Assessment]

The manuscript represents a **valuable** conceptual and technical contribution to our understanding of ciliogenesis and intraflagellar transport in vertebrates. Through a series of **solid** and technically superb live imaging experiments to directly visualize intraflagellar transport in various zebrafish ciliated tissues, the authors unveil the surprising breadth of intraflagellar transport speed among differing organs and link this to cell type-specific differences in cilia length and intraflagellar transport train size. This work will be of broad interest to researchers in numerous fields, including development, cell biology, and imaging.

---

## [Referee Report · Reviewer #2 (Public review)]

Summary:

In this study, the authors study intraflagellar transport (IFT) in cilia of diverse organs in zebrafish. They elucidate that IFT88-GFP (an IFT-B core complex protein) can substitute for endogenous IFT88 in promoting ciliogenesis and use it as a reporter to visualize IFT dynamics in living zebrafish embryo. They observe striking differences in cilia lengths and velocity of IFT trains in different cilia types, with smaller cilia length correlating with lower IFT speed. They generate several mutants and show that disrupting function of different kinesin-2 motors and BBSome or altering post translational modifications of tubulin does not have a significant impact on IFT velocity. They however observe that when the amount of IFT88 is reduced it impacts the cilia length, IFT velocity as well as the number and size of IFT trains. They also show that IFT train size is slightly smaller in one of the organs with shorter cilia (spinal cord). Based on their observations they propose that IFT velocity determines cilia length and go one step further to propose that IFT velocity is regulated by the size of IFT trains.

Strengths:

The main highlight of this study is the direct visualization of IFT dynamics in multiple organs of a living complex multi-cellular organism, zebrafish. The quality of the imaging is really good. Further, the authors have developed phenomenal resources to study IFT in zebrafish which would allow us to explore several mechanisms involved in IFT regulation in future studies. They make some interesting findings in mutants with disrupted function of kinesin-2, BBSome and tubulin modifying enzymes which are interesting to compare with cilia studies in other model organisms. Also, there observation of a possible link between cilia length and IFT speed is potentially fascinating.

Weaknesses:

The central hypothesis of the manuscript, which is cilia length regulation occurs via controlling IFT speed through the modulation of the size of the IFT complex, is supported only with preliminary data and needs stronger evidence.

The authors have robustly shown that the cilia length and IFT train speeds are highly variable between organs and have a strong correlation. With this they hypothesize that IFT train speeds could play a role in determining ciliary length, which is an interesting hypothesis that merits discussion. However, the claim that the cilia length (and IFT velocity) in different organs is different due to difference in the sizes of IFT trains is based on weak evidence. This is based on a marginal difference of IFT train sizes they observe between cilia of crista and spinal cord in immunofluorescence experiments (Fig. 5C). Inferring that this minor difference is key to the striking difference in cilia length and IFT velocity is too bold in my opinion.

To back this hypothesis, they look at ift88 morphants where there is a reduced pool of IFT88 (part of the IFTB1 complex which forms the core of IFT trains, based on multiple cryo-EM studies of IFT trains). Disruption (or reduced number) of IFTB1 complex could indeed lead to IFT trains not being formed properly, which can have an impact on IFT (train size, speed, frequency, etc.) and ciliary structure, as shown by the authors. However, this does not directly imply that under wild-type conditions, cilia in spinal cord have poorly formed slightly shorter IFT trains (cilia length ˜0.9 µm in spinal cord vs ˜1.2 µm in cristae; Fig. 3G) which results in strikingly lower speeds (˜0.4 µm/s in spinal cord vs ˜1.6 µm/s in cristae; Fig. 3G) and shorter cilia (˜3µm in spinal cord vs ˜26µm in cristae; Fig. 3H). Such a claim would require much stronger evidence.

Finally, if IFT train speeds directly correlate with size of IFT train, the authors should be able to see this within the same cilia, i.e., the velocity of a brighter IFT train (larger train) would be higher than the velocity of a dimmer IFT train (smaller train) within the same cilia. This is not apparent from the movies and such a correlation should be verified to make their claim stronger.

Impact:

Overall, I think this work develops an exciting new multicellular model organism to study IFT mechanisms. Zebrafish is a vertebrate where we can perform genetic modifications with relative ease. This could be an ideal model to study not just the role of IFT in connection with ciliary function but also ciliopathies. Further, from an evolutionary perspective, it is fascinating to compare IFT mechanisms in zebrafish with unicellular protists like Chlamydomonas, simple multicellular organisms like *C. elegans* and primary mammalian cell cultures. Having said that, the central hypothesis of the manuscript in not backed with strong evidence and I would recommend the authors to not give too much weight on the hypothesis that IFT train velocity is determined by the size of IFT trains. Given the technological advancements made in this study, I think it is fine if it is a descriptive manuscript and doesn't necessarily need a breakthrough hypothesis based on the marginal correlation they observe.

---

## [Referee Report · Reviewer #3 (Public review)]

Summary:

An interesting feature of cilia in vertebrates and many, if not all, invertebrates is the striking heterogeneity of their lengths among different cell types. As mutations interfering with ciliary length usually impair ciliary functions, ciliary length appears to be tuned for proper ciliary functions in a given type of cells. Although ciliary length is known to be affected by multiple factors, including intraflagellar transport (IFT), a cilia-specific, train-like bidirectional transportation, and ciliary proteins regulating microtubule dynamics, how it is intrinsically controlled remains largely elusive.

In the manuscript, the authors addressed this question from the angle of IFT by using zebrafish as a model organism. They demonstrated that ectopically expressed Ift88-GFP induced by heat shock treatment was able to sustain the normal development of and the cilia formation in ovl-/- zebrafish that would otherwise be dead by 7 dpf and lack of cilia due to the lack of Ift88, a critical component of IFT-B complex, suggesting a full function of the exogenous protein. They next live imaged Ift88-GFP in wild-type zebrafish larvae to visualize the IFT. Interestingly, they found that both anterograde and retrograde velocities of Ift88-GFP puncta differed in cilia of different cell types (crista, neuromast, pronephric duct, spinal chord, and epidermal cells) and displayed a positive correlation with the inherent length of the cilia. Similar results were obtained with ectopically expressed tdTomato-Ift43 driven by a beta-actin promoter. In the same cell type, however, the velocities of Ift88-GFP puncta did not alter in cilia of different lengths or at different developmental stages. Depletion of proteins such as Bbs4, Ttll3, Ttll6, and Ccp5 did not substantially alter the IFT velocities, excluding contributions of the BBSome or the enzymes involved in tubulin glycylation or glutamylation. They also used a cilia-localized ATP reporter to exclude the possibility of different ciliary ATP concentrations. When they compared the size of Ift88-GFP puncta in crista cilia, which are inherently long, and spinal chord cilia, which are relatively short, by imaging with a STED super-resolution microscope, they noticed a positive correlation between the puncta size, which presumably reflected the size of IFT trains, and the length of the cilia. Furthermore, in morphant larvae with slightly decreased Ift88 levels, judged by the grossly normal body axis, IFT particle sizes, their velocities, and ciliary lengths were all reduced as compared to control morphants. Therefore, they proposed that longer IFT trains facilitate faster IFT to result in longer cilia.

Strengths:

The authors demonstrated that: (1) both anterograde and retrograde IFT velocities can differ markedly in cilia of different cell types in zebrafish larvae; (2) specific IFT velocities are intrinsic to cell types; (3) IFT velocities in different types of cells are positively correlated with inherent ciliary lengths; and (4) IFT velocities are positively correlated with the size of IFT trains. These findings provide both new knowledge on IFT properties in zebrafish and insights that would facilitate understandings on mechanisms underlying the diversity of ciliary lengths in multicellular organisms. The experiments were carefully done and results are generally convincing. The imaging methods for tracing IFT in cilia of multiple cell types in zebrafish larvae are expected to be useful to other researchers in the field.

Weaknesses:

(1) Although the proposed model is reasonable, it is largely based on correlations.

(2) The effects of anti-sense RNA-induced Ift88 downregulation on IFT and ciliary length are artificial. It is unclear whether the levels of one or more IFT components are indeed regulated to control IFT train sizes and ciliary lengths in physiological conditions. Similarly, whether IFT velocities are indeed dictated by the size of IFT trains remains to be clarified.

(3) In the Discussion section, Kif17 is described as an important motor for IFT in mouse olfactory cilia. In the cited literature (Williams et al., 2014), however, Kif17 is reported to be dispensable for IFT in mouse olfactory cilia. This makes the discussions on Kif17 absurd.

---

## [Author Response]

The following is the authors’ response to the original reviews.

**Reviewer #1 (Public Review):**
(1) The main hypothesis/conclusion is summarized in the abstract: "Our study presents an intriguing model of cilia length regulation via controlling IFT speed through the modulation of the size of the IFT complex." The data clearly document the remarkable correlation between IFT velocity and ciliary length in the different cells/tissues/organs analyzed. The experimental test of this idea, i.e., the knock-down of GFP-IFT88, further supports the conclusion but needs to be interpreted more carefully. While IFT particle size and train velocity were reduced in the IFT88 morphants, the number of IFT particles is even more decreased. Thus, the contributions of the reduction in train size and velocity to ciliary length are, in my opinion, not unambiguous. Also, the concept that larger trains move faster, likely because they dock more motors and/or better coordinating kinesin-2 and that faster IFT causes cilia to be longer, is to my knowledge, not further supported by observations in other systems (see below).

Thank you for your comments. We agree with the reviewer that the final section on IFT train size, velocity, and ciliary length regulation requires additional evidence. The purpose of the knockdown experiments was to investigate the potential relationship between IFT speed and IFT train size. We hypothesize that a deficiency in IFT88 proteins may disrupt the regular assembly of IFT particles, leading to the formation of shorter IFT trains. Indeed, we observed a shorter IFT particles and slight reduction in the transport speed of IFT particles in the morphants. Certainly, it would be more convincing to distinguish these IFT trains through ultrastructural analysis. However, with current techniques, performing such analysis on the zebrafish model will be very difficult due to the limited sample size. In the revised version, we have tempered the conclusions in these sections, as suggested by other reviewers as well.

(2) I think the manuscript would be strengthened if the IFT frequency would also be analyzed in the five types of cilia. This could be done based on the existing kymographs from the spinning disk videos. As mentioned above, transport frequency in addition to train size and velocity is an important part of estimating the total number of IFT particles, which bind the actual cargoes, entering/moving in cilia.

Thank you. We have analyzed the entry frequency of IFT in five types of cilia, both anterior and posterior. The analysis indicates that longer cilia also exhibit a higher frequency of fluorescent particles entering the cilia. These results are presented in Figure 3J.

(3) Here, the variation in IFT velocity in cilia of different lengths within one species is documented - the results document a remarkable correlation between IFT velocity and ciliary length. These data need to be compared to observations from the literature. For example, the velocity of IFT in the quite long (~ 100 um) olfactory cilia of mice is similar to that observed in the rather short cilia of fibroblasts (~0.6 um/s). In Chlamydomonas, IFT velocity is not different in long flagella mutants compared to controls. Probably data are also available for *C. elegans* or other systems. Discussing these data would provide a broader perspective on the applicability of the model outside of zebrafish.

Thank you for your suggestions. We believe the most significant novelty of our manuscript is the discovery that IFT velocities are closely related to cilia length in an *in vivo* model system. Our data suggest that longer cilia may require faster IFT transport to maintain their stable length, powered by larger IFT trains. We did observe substantial variability in IFT velocities across different studies. For example, anterograde IFT transport ranges from 0.2 µm/s in mouse olfactory neurons (Williams *et al*, 2014) to 0.8 µm/s in 293T cells (See *et al*, 2016) and 0.4 µm/s in IMCD-3 cells (Broekhuis *et al*, 2014). Even in NIH-3T3 cells, two studies report significant differences, despite using the same IFT reporters: 0.3 µm/s versus 0.9 µm/s (Kunova Bosakova *et al*, 2018; Luo *et al*, 2017). These findings suggest that cell types and culture conditions can influence IFT velocities *in vitro*, which may not accurately represent *in vivo* conditions. Interestingly, research on mouse olfactory neurons showed a strong correlation between anterograde and retrograde IFT velocities. Additionally, IFT velocity is closely related to the cell types within the olfactory neuron population, consistent with our results (Williams *et al.*, 2014).

**Reviewer #2 (Public Review):**
Summary:In this study, the authors study intraflagellar transport (IFT) in cilia of diverse organs in zebrafish. They elucidate that IFT88-GFP (an IFT-B core complex protein) can substitute for endogenous IFT88 in promoting ciliogenesis and use it as a reporter to visualize IFT dynamics in living zebrafish embryos. They observe striking differences in cilia lengths and velocity of IFT trains in different cilia types, with smaller cilia lengths correlating with lower IFT speed. They generate several mutants and show that disrupting the function of different kinesin-2 motors and BBSome or altering post-translational modifications of tubulin does not have a significant impact on IFT velocity. They however observe that when the amount of IFT88 is reduced it impacts the cilia length, IFT velocity as well as the number and size of IFT trains. They also show that the IFT train size is slightly smaller in one of the organs with shorter cilia (spinal cord). Based on their observations they propose that IFT velocity determines cilia length and go one step further to propose that IFT velocity is regulated by the size of IFT trains.Strengths:The main highlight of this study is the direct visualization of IFT dynamics in multiple organs of a living complex multi-cellular organism, zebrafish. The quality of the imaging is really good. Further, the authors have developed phenomenal resources to study IFT in zebrafish which would allow us to explore several mechanisms involved in IFT regulation in future studies. They make some interesting findings in mutants with disrupted function of kinesin-2, BBSome, and tubulin modifying enzymes which are interesting to compare with cilia studies in other model organisms. Also, their observation of a possible link between cilia length and IFT speed is potentially fascinating.Weaknesses:The manuscript as it stands, has several issues.(1) The study does not provide a qualitative description of cilia organization in different cell types, the cilia length variation within the same organ, and IFT dynamics. The methodology is also described minimally and must be detailed with more care such that similar studies can be done in other laboratories.

Thank you for your comments. We found that cilia length is generally consistent within the same cell types we examined, including those in the pronephric duct, spinal cord, and epidermal cells. However, we observed variability in cilia length within ear crista cilia. Upon comparing IFT velocities, we found no differences among these cilia, further confirming our conclusion that IFT velocity is directly related to cell type rather than cilia length. These new results are presented in Figure S4 of the revised version.

We apologize for the lack of methodological details in the original manuscript. Following the reviewer's suggestion, we have added a detailed description of the methods used to generate the transgenic line and to perform IFT velocity analysis. These details are included in Figure S2 and are thoroughly described in the methods section of the revised manuscript.

(2) They provide remarkable new observations for all the mutants. However, discussion regarding what the findings imply and how these observations align (or contradict) with what has been observed in cilia studies in other organisms is incomprehensive.

Thank you for this suggestion. We initially submitted this paper as a report, which have word limits. We believe the main finding of our work is that IFT velocity is directly associated with cell type, with longer cilia requiring higher velocities to maintain their length. This association of IFT velocity with cell type has also been observed in mouse olfactory neurons(Williams *et al.*, 2014). We have included a discussion of our findings, along with related data published in other organisms, in the revised version.

(3) The analysis of IFT velocities, the main parameter they compare between experiments, is not described at all. The IFT velocities appear variable in several kymographs (and movies) and are visually difficult to see in shorter cilia. It is unclear how they make sure that the velocity readout is robust. Perhaps, a more automated approach is necessary to obtain more precise velocity estimates.

Thank you for these comments. To measure the IFT velocities, we first used ImageJ software to generate a kymograph, where moving particles appear as oblique lines. The velocity of these particles can be calculated based on the slope of the lines (Zhou *et al*, 2001). In the initial version, most of the lines were drawn manually. To eliminate potential artifacts, we also used KymographDirect software to automatically trace the particle paths. The velocities obtained with this method were similar to those calculated manually. These new data are now shown in Figure S2 B-D. For shorter cilia, we only used particles with clear moving paths for our calculations. In the revised version, we have included a detailed description of the velocity analysis methods.

(4) They claim that IFT speeds are determined by the size of IFT trains, based on their observations in samples with a reduced amount of IFT88. If this was indeed the case, the velocity of a brighter IFT train (larger train) would be higher than the velocity of a dimmer IFT train (smaller train) within the same cilia. This is not apparent from the movies and such a correlation should be verified to make their claim stronger.

Thank you for these excellent suggestions. We measured the particle size and fluorescence intensity of 3 dpf crista cilia using high-resolution images acquired with Abberior STEDYCON. The results showed a positive correlation between the two. These data have been added to the revised version in Figure 5I, which includes both control and ift88 morphant data.

(5) They make an even larger claim that the cilia length (and IFT velocity) in different organs is different due to differences in the sizes of IFT trains. This is based on a marginal difference they observe between the cilia of crista and the spinal cord in immunofluorescence experiments (Figure 5C). Inferring that this minor difference is key to the striking difference in cilia length and IFT velocity is incorrect in my opinion.Impact:Overall, I think this work develops an exciting new multicellular model organism to study IFT mechanisms. Zebrafish is a vertebrate where we can perform genetic modifications with relative ease. This could be an ideal model to study not just the role of IFT in connection with ciliary function but also ciliopathies. Further, from an evolutionary perspective, it is fascinating to compare IFT mechanisms in zebrafish with unicellular protists like Chlamydomonas, simple multicellular organisms like C elegans, and primary mammalian cell cultures. Having said that, the underlying storyline of this study is flawed in my opinion and I would recommend the authors to report the striking findings and methodology in more detail while significantly toning down their proposed hypothesis on ciliary length regulation. Given the technological advancements made in this study, I think it is fine if it is a descriptive manuscript and doesn't necessarily need a breakthrough hypothesis based on preliminary evidence.

Thanks for with these comments. We agree with this reviewer that more evidences are required to explain why IFT is transported faster in longer cilia. In the revised version, we have modified and softened this section, focusing primarily on the novel findings of IFT velocity differences between cilia of varying lengths.

**Reviewer #3 (Public Review):**
Summary:A known feature of cilia in vertebrates and many, if not all, invertebrates is the striking heterogeneity of their lengths among different cell types. The underlying mechanisms, however, remain largely elusive. In the manuscript, the authors addressed this question from the angle of intraflagellar transport (IFT), a cilia-specific bidirectional transportation machinery essential to biogenesis, homeostasis, and functions of cilia, by using zebrafish as a model organism. They conducted a series of experiments and proposed an interesting mechanism. Furthermore, they achieved in situ live imaging of IFT in zebrafish larvae, which is a technical advance in the field.Strengths:The authors initially demonstrated that ectopically expressed Ift88-GFP through a certain heatshock induction protocol fully sustained the normal development of mutant zebrafish that would otherwise be dead by 7 dpf due to the lack of this critical component of IFT-B complex.Accordingly, cilia formations were also fully restored in the tissues examined. By imaging the IFT using Ift88-GFP in the mutant fish as a marker, they unexpectedly found that both anterograde and retrograde velocities of IFT trains varied among cilia of different cell types and appeared to be positively correlated with the length of the cilia.For insights into the possible cause(s) of the heterogeneity in IFT velocities, the authors assessed the effects of IFT kinesin Kif3b and Kif17, BBSome, and glycylation or glutamylation of axonemal tubulin on IFT and excluded their contributions. They also used a cilia-localized ATP reporter to exclude the possibility of different ciliary ATP concentrations. When they compared the size of Ift88-GFP puncta in crista cilia, which are long, and spinal cord cilia, which are relatively short, by imaging with a cutting-edge super-resolution microscope, they noticed a positive correlation between the puncta size, which presumably reflected the size of IFT trains, and the length of the cilia.Finally, they investigated whether it is the size of IFT trains that dictates the ciliary length. They injected a low dose (0.5 ng/embryo) of ift88 MO and showed that, although such a dosage did not induce the body curvature of the zebrafish larvae, crista cilia were shorter and contained less Ift88-GFP puncta. The particle size was also reduced. These data collectively suggested mildly downregulated expression levels of Ift88-GFP. Surprisingly, they observed significant reductions in both retrograde and anterograde IFT velocities. Therefore, they proposed that longer IFT trains would facilitate faster IFT and result in longer cilia.Weaknesses:The current manuscript, however, contains serious flaws that markedly limit the credibility of major results and findings. Firstly, important experimental information is frequently missing, including (but not limited to) developmental stages of zebrafish larvae assayed (Figures 1, 3, and 5), how the embryos or larvae were treated to express Ift88-GFP (Figures 3-5), and descriptions on sample sizes and the number of independent experiments or larvae examined in statistical results (Figures 3-5, S3, S6). For instance, although Figure 1B appears to be the standard experimental scheme, the authors provided results from 30-hpf larvae (Figure 3) that, according to Figure 1B, are supposed to neither express Ift88-GFP nor be genotyped because both the first round of heat shock treatment and the genotyping were arranged at 48 hpf. Similarly, the results that ovl larvae containing Tg(hsp70l:ift88 GFP) (again, because the genotype is not disclosed in the manuscript, one can only deduce) display normal body curvature at 2 dpf after the injection of 0.5 ng of ift88 MO (Fig 5D) is quite confusing because the larvae should also have been negative for Ift88-GFP and thus displayed body curvature. Secondly, some inferences are more or less logically flawed. The authors tend to use negative results on specific assays to exclude all possibilities. For instance, the negative results in Figures 4A-B are not sufficient to "suggest that the variability in IFT speeds among different cilia cannot be attributed to the use of different motor proteins" because the authors have not checked dynein-2 and other IFT kinesins. In fact, in their previous publication (Zhao et al., 2012), the authors actually demonstrated that different IFT kinesins have different effects on ciliogenesis and ciliary length in different tissues. Furthermore, instead of also examining cilia affected by Kif3b or Kif17 mutation, they only examined crista cilia, which are not sensitive to the mutations. Similarly, their results in Figures 4C-G only excluded the importance of tubulin glycylation or glutamylation in IFT. Thirdly, the conclusive model is based on certain assumptions, e.g., constant IFT velocities in a given cell type. The authors, however, do not discuss other possibilities.

Thank you for pointing out the flaws in our experiments. We apologize for any confusion caused by the lack of detail in our descriptions. Regarding Figure 2B, we want to clarify that it depicts the procedure for heat shock experiments conducted for the *ovl* mutants' rescue assay, not the experimental procedure for IFT imaging. In the revised version, we have included detailed methods on how to induce the expression of Ift88-GFP via heat shock and the subsequent image processing. The procedure for heat induction is also shown in Figure S2A. We have also added the sample sizes for each experiment and descriptions of the statistical tests used in the appropriate sections of the revised version.

Regarding the comments on the relationship between IFT speed variability and motor proteins, we completely agree with the reviewer. We have revised our description of this part accordingly.

Lastly, the results shown in Figure 5D are from a wild-type background, not *ovl* mutants. We aimed to demonstrate that a lower dose of *ift88* morpholino (0.5 ng) can partially knock down Ift88, allowing embryos to maintain a generally normal body axis, while the cilia in the ear crista became significantly shorter.

**Recommendations for the authors:**

**Reviewer #1 (Recommendations For The Authors):**
Minor(I recommend adding page numbers and probably line numbers. This makes commenting easier)

We have added page numbers and line numbers in the revised manuscript.

Intro: Furthermore, ultra-high-resolution microscopy showed a close association between cilia length in different organs and the size of IFT fluorescent particles, indicating the presence of larger IFT trains in longer cilia.This correlation is not that strong and data are only available for 2 types of cilia.

Thanks. We have modified this part.

(P5) cilia (Fig. 1D) -> (Fig. S1)

Thanks. We have corrected this.

(P5) "These movies provide a great opportunity to compare IFT across different cilia." Rewrite: "This approach allows one to determine the velocity and frequency based of IFT based on kymographs" or similar.

Thank you for your correction, we have changed it in the revised manuscript.

This observation suggests that cargo and motor proteins are more effectively coordinated in transporting materials, resulting in increased IFT velocity-a novel regulatory mechanism governing IFT speed in vertebrate cilia.This is a somewhat cryptic phrase, rewrite?

We have modified this sentence.

P6 and elsewhere: "IFT in the absence of Kif17 or Bbs proteins" I wonder if it would be better to provide subheadings summarizing the main observation instead of descriptive titles. This includes the title of the manuscript.

Thanks for this suggestion. We have changed the title of subheadings in the revised manuscript. We prefer to keep the current title of this manuscript, as we think this paper is mainly to describe IFT in different types of cilia.

Is it known whether IFT protein and motors are alternatively spliced in the various ciliated cells of zebrafish? In this context, is it known whether the cells express IFT proteins at different levels?

We analyzed the transcript isoforms of several ciliary genes, including *ift88, ift52, ift70, ift172,* and *kif3a*. Most of these IFT genes possess only a single transcript isoform. The Kif3a motor proteins have two isoforms (long and short isoforms), however, the shorter isoform contains only the motor domain and is presumed to be nonfunctional for IFT. While we cannot completely rule out this possibility, we consider it unlikely that the variation in IFT speed is due to alternative splicing in ciliary tissues.

(P6) The relation between osm-3 and Kif17 needs to be introduced briefly.

Thank you for pointing this out. We have added it in the proper place of the revised manuscript.

(P6) "IFT was driven by kinesin or dynein motor proteins along the ciliary axoneme." "is driven"?Delete phrase and IFT to the next sentence?

We have deleted this sentence.

(P7) "Moreover, the mutants were able to survive to adulthood and there is no difference in the fertility or sperm motility between mutants and control siblings, which is slightly different from those observed in mouse mutants (Gadadhar et al., 2021)." Could some of these data be shown?

Thanks for this suggestion. When crossed with wild-type females, all homozygous mutants showed no difference in fertility compared to controls. The percentage of fertilization rates in mutants was 90.5% (n = 7), which was similar to wild-type (87.2%, n = 7). We determined the trajectories of free-swimming sperm by high-speed video microscopy. The vast majority of sperm in *ttll3* mutant, similar to wild-type sperm, swim almost entirely along a straight path, which is different from what was observed in the mouse mutant (where 86% of *TTLL3*-/-*TTLL8*-/- sperm rotate in situ). We assessed cilia motility in the pronephric ducts of 5dpf embryos using high-speed video microscopy. The *ttll3* mutant exhibited a rhythmic sinusoidal wave pattern similar to the control, and there was no significant difference in ciliary beating frequency. These new data are now included in Figure S7C-H.

(P7) "which has been shown early to reduce" earlier

We have changed it. Thanks.

Maybe the authors could speculate how the cells ensure the assembly of larger/faster trains in certain cells. Are the relative expression levels known or worth exploring?

Thank you for these suggestions. We believe that longer cilia may maintain larger IFT particle pools in the basal body region, facilitating the assembly of large IFT trains. The higher frequency of IFT injection in longer cilia further supports this hypothesis. It is likely that cells with longer cilia have higher expression levels of IFT proteins. However, due to the lack of proper antibodies for IFT proteins in zebrafish, it is currently unfeasible to compare this. This experiment is certainly worth investigating in the future. We have added this discussion in the revised manuscript.

**Reviewer #2 (Recommendations for The Authors):**
Here are detailed comments for the authors:(1) The authors need to describe their methodology of imaging and what they observe in much greater detail. How were the different cilia types organized? Approximately how many were observed in every organ? How were they oriented? Were there length variations between cilia in the same organ? While imaging, were individual cilium mostly lying in a single focal plane of imaging or the authors often performed z-scans over multiple planes. Velocity measurement is highly variable if individual cilia are spanning over a large volume, with only part of it in focus in single plane acquisition.

Thank you for your comments. We apologize for the lack of details in the methodology. We have added a detailed description in the 'Materials and Methods' section and illustrated the experimental paradigm in Figure S2A of the revised manuscript. In most tissues we examined, the length of cilia was relatively uniform, except in the crista. The cilia in the crista were significantly longer, with lengths varying between 5 and 30 μm, compared to those in other tissues. We categorized the cilia lengths in the crista into three groups at intervals of 10 μm and measured the anterograde and retrograde velocities of IFT in each group. The results, shown in Figure S4, revealed no significant difference in IFT velocity among the different cilia lengths within the same tissue. Regarding the imaging, all IFT movies were captured in a single focal plane. In most cases, we did not observe significant velocity variability within the same cilium.

(2) It is very difficult to directly observe the large differences in IFT velocity from the kymographs, especially in the case of shorter cilia and retrograde motion in them. The quality of the example kymographs could be improved and more zoomed in several cases.

Thank you for this suggestion. We have modified this.

(3) The authors do not describe at all, how velocity analysis was done on the kymographs? Were lines drawn manually on the kymographs? From the movies and the kymographs it is visible that the IFT motion is often variable and sometimes gets stuck. How did the authors determine the velocities of such trains? A single slope through the entire train or part of the train? Were they consistent with this? Such variable motion is not so easy to discern in the case of really short cilia. The authors could use a more automatic way of extracting velocities from kymographs using tools such as kymodirect or kymobutler. Keeping in mind that IFT velocity is the main parameter studied in this work, it is important that the analysis is robust.

We apologize for the previous lack of detailed description. We utilized ImageJ software to generate kymographs, where particles appear as lines. For a moving particle, this line appears oblique. We manually drew lines on the kymographs, and the velocity of particles was calculated based on the slope (Zhou *et al.*, 2001). We only analyzed particles that tracked the full length of the cilia. Following the reviewer's suggestions, we also used the automatic software KymographDirect to calculate the velocity of IFT particles. The results were similar to those calculated using the previous method. These new data are now shown in Figure S2B-D. For shorter cilia, we only used particles with clear moving paths for our calculations. In the revised version, we have included a detailed description of the velocity analysis methods.

(4) In line with the previous point, as visible from the kymographs the velocity is significantly slower near the transition zone. Did the authors make sure they are not including the region around the transition zone while measuring the IFT velocity, especially in the case of shorter cilia?

Thank you for the comment. In the revised manuscript, we automatically extracted the path of particle using KymographDirect software. Quantification of each particle's velocity versus position in crista reveals that anterograde IFT proceeds from the base to the tip at a relatively constant speed, whereas retrograde IFT undergoes a slightly acceleration process when returning to the base (Fig. S2E). This finding differs from observations in *C. elegans*, which dynein-2 first accelerating and then decelerating back to 1.2 μm/s adjacent to the ciliary base (Yi *et al*, 2017). We believe it is very unlikely that the slow IFT velocity is due to the calculation of IFT only in the transition zone of shorter cilia.

(5) There are several fascinating findings in this work that the authors do not discuss properly. Firstly, do the authors have a hypothesis as to why IFT speeds are so radically different in different cilia types, given that they are driven by the same motor proteins and have the same ATP levels? They make a big claim in this paper that IFT train sizes correlate with train velocities. IFT trains have a highly ordered structure with regular binding sites for motor proteins. So, a smaller train would have a proportional number of motors attached to them. Why (and how) are the motors moving trains so slowly in some cilia and not in others? If there is no clear answer, the authors must put forward the open question with greater clarity.

Thank you for the comment. We hypothesize that if multiple motors drive the movement of cargoes synergistically, it could increase the speed of IFT transport. An example supporting this hypothesis is the principle of multiple-unit high-speed trains, which use multiple motors in each individual car to achieve high speeds. Of course, this is just one hypothesis, and we cannot exclude other possibilities, such as the use of different adaptors in different cell types. We have revised our conclusions accordingly in the updated manuscript.

(6) They find that IFT speeds do not change in kif17 mutants. Are the cilia length also similar (does not appear to be the case in Figure 4 and Figure S3)? Cilia length needs to be quantified. Further, they mention that in C elegans, heterotrimeric kinesin-2 and homodimeric kinesin-2 coordinate IFT. However, from several previous studies, we know that in Chlamydomonas and in mammalian cilia IFT is driven primarily by heterotrimeric kinesin-2 with no evidence that homodimeric kinesin-2 is linked with driving IFT. It appears to be the same in zebrafish. This is an interesting finding and needs to be discussed far more comprehensively.

Thank you for your comments. We have previously shown that the number and length of crista cilia were grossly normal in *kif17* mutants (Zhao *et al*, 2012). The length of crista cilia displayed slight variability even in wild-type larvae. We quantified the length of cilia in both the crista and neuromast within different mutants, and our analysis revealed no significant difference (see Author response image 1). We agree with the reviewer that Kif17 may play a minor role in driving IFT in cilia. However, previous studies have shown that KIF17 exhibits robust, processive particle movement in both the anterograde and retrograde directions along the entire olfactory sensory neuron cilia in mice. This suggests that, although not essential, KIF17 may also be involved in IFT (Williams *et al.*, 2014). We have added more discussion about Kif17 and heterotrimeric kinesin in the appropriate section of the revised manuscript.

**Author response image 1. sa3fig1:** Statistical significance is based on Kruskal-Wallis statistic, Dunn's multiple comparisons test. n.s., not significant, p＞0.05.

(7) Again, they find that IFT speeds do not change in BBS-4 mutants. I have the same comment about the cilia length as for kif17 mutants. Further, the discussion for this finding is lacking. The authors mention that IFT is disrupted in BBSome mutants of C elegans. Is this the case in other organisms as well? Structural studies on IFT trains reveal that BBSomes are not part of the core structure, while other studies reveal that BBSomes are not essential for IFT. So perhaps the results here are not too surprising.

We agree with the reviewer that BBSome is possibly not essential for IFT in most cilia. However, in the cilia of olfactory sensory neurons, BBSome is involved in IFT in both mice and nematodes (Ou *et al*, 2005; Williams *et al.*, 2014). We have added more discussion about BBSome in the appropriate section of the revised manuscript.

(8) No change in IFT velocities in kif3b mutants is rather surprising. The authors suggest that Kif3C homodimerizes to carry out IFT in the absence of Kif3B. Even if that is the case, the individual homodimer constituents of heterotrimeric kinesin-2 have been shown in previous studies to have different motor properties when homodimerized artificially. Why is IFT not affected in these mutants? This should be discussed. Also, the cilia lengths should be quantified.

We think the presence of the Kif3A/Kif3C/KAP3 trimeric kinesin may substitute for the Kif3A/Kif3B/KAP3 motors in *kif3b* mutants, which show normal length of cristae cilia. The Kif3A/Kif3C/KAP3 trimeric kinesin may have similar transport speeds as the Kif3A/Kif3B/KAP3 motors. We did not propose that the Kif3C homodimer can drive the cargoes alone. We apologize for this misunderstanding. Additionally, we have reevaluated the IFT velocities among different lengths of cristae cilia and found no difference between longer and shorter cilia within the same cell types.

(9) The findings with tubulin modifications should also be discussed in comparison to what has been observed in other organisms.

We have added further discussion about this result in the revised manuscript.

(10) The authors find that IFT velocity is lower in ift88 morphants. They also find that the cilia length is shorter (in which cilia type?). Immunofluorescence experiments show that the IFT particle number and size are lower in the ift88 morphants. How many organisms did they look at for this data? What is the experimental variability in intensity measurements in immunofluorescence experiments? Wouldn't the authors expect much higher variability in ift88 morphants (between individual organisms) due to different amounts of IFT88 than for wildtype?

Thank you for your comments. We apologize for the lack of information regarding the number of organisms observed in Figure 5. These numbers have been added to the figure legends in the revised manuscript. When a low dose of *ift88* morpholino was injected, we observed significant shortening of cilia in the ear crista, along with reduced IFT speed. We measured the fluorescence intensity of different IFT particles and found a positive correlation between IFT particle size and fluorescence intensity (Fig 5I). Moreover, the variability of cilia length in cristae is slightly higher in ift88 morphants. These new data have been included in the revised version.

(11) From their observations they make the claim that IFT velocity is directly proportional to IFT train size. Now within every cilium, IFT trains have large size variations, given the variable intensities for different IFT trains. The authors themselves show that they resolve far more trains when imaging with STED (possibly because they are able to visualize the smaller trains). Is the IFT velocity within the same cilium directly correlated with the intensity of the train, both for wildtype and ift88 morphants? That is the most direct way the authors can test that their hypothesis is true. Higher intensity (larger train size) results in faster velocity. From a qualitative look at their movies, I do not see any strong evidence for that.

Thank you for your comments. We have measured the particle size and fluorescence intensity of 3dpf crista cilia using high-resolution images acquired with Abberior STEDYCON. The results, shown in Figure 5I, demonstrate a positive correlation between particle size and fluorescence intensity.

(12) Are the sizes of both anterograde and retrograde trains lower in ift88 morphants? It's not clear from the data. It should be clearly stated that the authors speculate this and this is not directly evident from the data.

Because the size of IFT fluorescence particles is based on immunostaining results, not live imaging, we cannot determine whether they are anterograde or retrograde IFT particles.

Therefore, we can only speculate that possibly both anterograde and retrograde trains are reduced in *ift88* morphants.

(13) The biggest claim in this paper is that the cilia lengths in different organs are different due to differences in IFT train sizes. This is based on highly preliminary data shown in Figure 5C (how many organisms did they measure?). The difference is marginal and the dataset for spinal cord cilia is really small. The internal variability within the same cilia type is larger than the difference. How is this tiny difference resulting in such a large difference in IFT speeds? I believe their conclusions based on this data are incorrect.

From our results, we believe that IFT velocity is related to cell types rather than the length of cilia (Fig. S4), which has also been mentioned in previous studies (Williams *et al.*, 2014). We agree with the reviewer that the evidence for faster IFT speed due to larger train size is not very solid. We have accordingly softened our conclusion and mentioned other possibilities in the revised version.

Minor comments:(1) The authors only mention the number of IFT particles for their data. They should provide the number of cilia and the number of organisms as well.

Thank you for your suggestion. We added the number of cilia and organisms next to the number of particles in Figure 3, Figure S2-S5 and Table S1 of the revised manuscript.

(2) Cilia and flagella are similar structurally but not the same. The authors should change the following sentence: In contrast to the localization of most organelles within cells, cilia (also known as flagellar) are microtubule-based structures that extend from the cell surface, facilitating a more straightforward quantification of their size.

Thank you for the detailed review. We have changed it in our revised manuscript.

(3) The authors should provide references here. For example, Chlamydomonas has two flagella with lengths ranging from 10 to 14 μm, while sensory cilia in *C. elegans* vary from approximately 1.5 μm to 7.5 μm. In most mammalian cells, the primary cilium typically measures between 3 and 10 μm.

We have added it in our revised manuscript.

(4) They should mention ovl mutants are IFT88 mutants when they introduce it in the main text.

We have added it in our revised manuscript.

(5) Correct the grammar here: The velocity of IFT within different cilia also seems unchanged (Figure 4F, Movie S9, Table S1).

We have changed it.

(6) Correct the grammar here: Similarly, the IFT speeds also exhibited only slight changes in ccp5 morphants, which decreased the deglutamylase activities of Ccp5 and resulted in a hyperglutamylated tubulin

We have changed it.

**Reviewer #3 (Recommendations For The Authors):**
Introduction:1st paragraph, "flagellar" should be "flagella"; 2nd paragraph, "result a wide range of" should be "result in a...".

We have changed it.

Results and discussion:"...certain specialized cell types, including olfactory epithelia and pronephric duct, ...": olfactory epithelia and pronephric duct are tissues, not cells."...the GFP fluorescence of the transgene was prominently enriched in the cilia (Fig 1D)" : Fig 2D?"The velocity of IFT within different cilia was also seems unchanged (Fig. 4 F, Movie S9, Table S1)": "was" and "seems" cannot be used together."...driven by b-actin2 promotor": -actin2?"...each dynein motor protein might propel multiple IFT complexes": The "protein" should be deleted.

Thanks. We have corrected all of these mistakes.

Figures:Figure 1: Dyes and antibodies used other than the anti-acetylated tubulin antibody should mentioned. The developmental stages of zebrafish used for the imaging are mostly missing.

Thanks. In the revised version, we have updated the figure legends to include descriptions of the antibodies, developmental stages, as well as N numbers.

Figure 2B: What "hphs" means should be explained somewhere.

Thanks. We have added full name for these abbreviations.

Figures 3A-E: For clarity, the cilia whose IFT kymographs are shown should be marked. "Representative particle traces are marked with white lines in panels D and E" (legend): they are actually black lines. The authors should also clearly disclose the developmental stages of zebrafish used for the imaging.

Thank you for your comments. In the revised manuscript, the cilia used to generate the kymograph are marked by yellow arrows. We have updated the legend to change "white" to "black." Additionally, we have included the developmental stages of zebrafish used for imaging in Figure 3A.

Figures 3G-K: The authors used quantification results from 4-dpf larvae and 30-hpf embryos for comparisons. Nevertheless, according to their experimental scheme in Figure 2B, 30-hpf embryos were not subjected to heat-shock treatment and genotyping. How could they express Ift88-GFP for the imaging? How could the authors choose larvae of the right genotypes? In addition, even if the authors heat-shocked them in time but forgot to mention, there are issues that need to be clarified experimentally and/or through citations, at least through discussions. Firstly, at 30 hpf, those motile cilia are probably still elongating. If this is the case, their final lengths would be longer than those presented (H; the authors need to disclose whether the lengths were measured from ciliary Ift88-GFP or another marker). In other words, the correlation with IFT velocities (H and I) might no longer exist when mature cilia were measured. Similarly, cilia undergo gradual disassembly during the cell cycle. Epidermal cells at 30-hpf are likely proliferating actively, and the average length of their cilia (H) would be shorter than that measured from quiescent epidermal cells in later stages.

Thank you for these comments. First, we want to clarify that Figure 2B depicts the procedure for heat shock experiments conducted for the ovl mutants' rescue assay, not the experimental procedure for IFT imaging. We visualized IFT in five types of cilia using *Tg (hsp70l: ift88-GFP)* embryos without the *ovl* mutant background. In the revised manuscript, we have provided a detailed description of embryo treatment in the 'Materials and Methods' section and illustrated the experimental paradigm in Figure S2A.

Regarding the ciliary length differences between different developmental stages, we quantified cilia length in epidermal cells at 30 hpf versus 4 dpf, and in pronephric duct cilia at 30 hpf versus 48 hpf. Our analysis found no significant difference in length between earlier and later stages. Additionally, IFT velocities were comparable between these stages. These findings suggest that slower IFT velocities may not be attributed to the selection of different embryonic stages. Furthermore, we demonstrated that longer and shorter cilia maintain similar IFT velocities in crista cilia, indicating that elongated cilia within the same cell type exhibit comparable IFT velocities. These new results are presented in Figures S4 and S5 in the revised version.

Secondly, do IFT velocities differ between elongating and mature cilia or remain relatively constant for a given cell type? The authors apparently take the latter for granted without even discussing the possibility of the former. In addition, whether the quantification results were from cilia of one or multiple fish, an important parameter to reflect the reproducibility, and sample sizes for the length data are not disclosed. The lack of descriptions on sample sizes and the number of independent experiments or larvae examined are actually common for statistical results in this manuscript.

Thank you for your comments. We apologize for omitting the basic description of sample sizes and the number of cilia analyzed. We have addressed these issues in the revised manuscript. The length of 4dpf Crista cilia is variable, with longer cilia reaching up to 30 µm and shorter cilia measuring only around 5 µm within the same crista. We categorized the cilia length of Crista into three groups at intervals of 10 µm and measured anterograde and retrograde velocities of IFT in each group. The results revealed no significant difference in IFT velocity among elongating and mature cilia within crista. These supplementary data are now included in Figure S4.

Figures 4A-B: When mutating neither Kif17 nor Kif3b affected the IFT of crista cilia, the data unlikely "suggest that the variability in IFT speeds among different cilia cannot be attributed to the use of different motor proteins". In fact, in the cited publication (Zhao et al., 2012), the authors used the same and additional mutants (Kif3c and Kif3cl) to demonstrate that different IFT-related kinesin motors have different effects on ciliogenesis and ciliary length in different tissues, results actually implying tissue-specific contributions of different kinesin motors to IFT. Furthermore, although likely only cytoplasmic dynein-2 is involved in the retrograde IFT, the authors cannot exclude the possibility that different combinations or isoforms of its many subunits and regulators contribute to the velocity regulation. Therefore, the authors need to reconsider their wording. This reviewer would suggest that the authors examine the IFT status of cilia that were previously reported to be shortened in the Kif3b mutant to see whether the correlation between ciliary length and IFT velocities still stands. This would actually be a critical assay to assess whether the proposed correlation is only a coincidence or indeed has a certain causality.

Thank you for your comments. The shortened cilia observed in Kif3b mutants may be attributed to the presence of maternal Kif3b proteins, making it challenging to exclude the involvement of Kif3b motor. Regarding the relationship between IFT speed variability and motor proteins, we agree with the reviewer that we cannot entirely dismiss the possibility of different motors or adaptors being involved. We have revised our description of this aspect accordingly.

Figures 4C-G: Similarly, when the authors found that tubulin glycylation or glutamylation has little effect on IFT, they cannot use these observations to exclude possible influences of other types of tubulin modifications on IFT. They should only stick to their observations.

Yes, we agree. We have changed the description in the revised manuscript.

Figure 5:A-C: When the authors only compared immotile cilia of crista with motile cilia of the spinal cord, it is hard to say whether the difference in particle size is correlated with ciliary length or motility. Cilia from more tissues should be included to strengthen their point, especially when the authors want to make this point the central one.D: The authors showed that ovl larvae containing Tg(hsp70l:ift88 GFP) (as they do not indicate the genotype, this reviewer can only deduce) display normal body curvature at 2 dpf after the injection of 0.5 ng of ift88 MO. Such a result, however, is quite confusing. According to their experimental scheme in Figure 2B, these larvae were not subjected to heat shock induction for Ift88-GFP. Do ovl larvae containing Tg(hsp70l:ift88 GFP) naturally display normal body curvature at 2 dpf?

Thank you for your comments. Due to technical limitations, comparing IFT particle size across different cilia using STED is challenging. We agree with this reviewer that the evidence supporting this aspect is relatively weak. Accordingly, we have modified and softened our conclusion in the revised version.

Regarding the injection of *ift88* morpholino, we want to clarify that we are injecting it into wildtype embryos, not *oval* mutants. The lower dose of *ift88* morpholino (0.5ng) partially knocked down Ift88, allowing embryos to maintain a grossly normal body axis while resulting in shorter cilia in the ear crista.

E: The authors need to indicate the developmental stage of the larvae examined. One piece of missing data is global expression levels of both endogenous (maternal) Ift88 and exogenousIft88-GFP in zebrafish larvae that are either uninjected, 8-ng-ift88 MO-injected, or 0.5-ng-ift88 MO-injected, preferably at multiple time points up to 3 dpf. The results will clarify (1) the total levels of Ift88 following time; (2) the extent of downregulation the MO injections achieved at different developmental stages; and importantly (3) whether the low MO dosage (0. 5 ng) indeed allowed a persistent downregulation to affect IFT trains at 3 dpf, a time the authors made the assays for Figures 5F-J to reach the model (K). It will be great to include wild-type larvae for comparison.

Thank you for these valuable suggestions. The *ift88* morpholino (MO) was designed to block the splicing of *ift88* transcripts and has been used in multiple studies. This morpholino specifically blocks the expression of endogenous *ift88*, while the expression of the Ift88-GFP transgene remains unaffected. It would be beneficial to titrate the expression level of Ift88 in the morphants at different stages. Unfortunately, we do not have access to a zebrafish Ift88 antibody. We assessed the effects of a lower amount of MO based on our observation that the fish maintained a normal body axis while exhibiting shorter cilia. Ideally, the amount of Ift88 should be lower in the morphants, considering the presence of ciliogenesis defects. We have included additional comments regarding this limitation in the revised version.

Movies:Movies 1-5: Elapsed time is not provided. Furthermore, cilia in the pronephric duct and spinal cord are known to beat rapidly. Their motilities, however, appear to be largely compromised in Movies 3 and 4. Although the quantification results in Fig 3G imply that the authors imaged 30hpf embryos for such cilia, there is no statement on real conditions.

Thank you for your comments. We apologize for missing elapsed time in our movies. We have addressed this issue in the revised manuscript. Motile cilia are difficult to image due to their fast beating. To immobilize the moving cilia and enable the capture of IFT movement within the cilia, we gently press the embryo with a round cover glass to inhibit the beating of cilia. Data from each embryo were collected within 5 minutes to avoid the impact of embryo death on the results. We have added detail description in the 'Materials and Methods' section.

Materials:The sequence of morpholino oligonucleotide against ift88 is missing.

We have added the sequence of *ift88* morpholino in the revised manuscript.

References:Important references are missing, including (1) the paper by Leventea et al., 2016 (PMID: 27263414), which shows cilia morphologies in various zebrafish tissues with more detailed descriptions of tissue anatomies and experimental techniques; (2) papers documenting that dynein motors "move faster than Kinesin motors" in IFT of C. reinhardtii and *C. elegans* cilia; and (3) the paper by Li et al., 2020 (PMID: 33112235), in which the authors constructed a hybrid IFT kinesin to markedly reduced anterograde IFT velocity (~ 2.8 fold) and IFT injection rate in C. reinhardtii cilia and found only a mild reduction (~15%) in ciliary length. This paper is important because it is a pioneer one that elegantly investigated the relationship between IFT velocity and ciliary length. The findings, however, do not necessarily contradict the current manuscript due to differences in, e.g., model organisms and methodology.

Thank you for the detailed review, we have cited these literatures in the proper place of the revised manuscript.

Reference

Broekhuis JR, Verhey KJ, Jansen G (2014) Regulation of cilium length and intraflagellar transport by the RCK-kinases ICK and MOK in renal epithelial cells. *PLoS One* 9: e108470

Kunova Bosakova M, Varecha M, Hampl M, Duran I, Nita A, Buchtova M, Dosedelova H, Machat R, Xie Y, Ni Z *et al* (2018) Regulation of ciliary function by fibroblast growth factor signaling identifies FGFR3-related disorders achondroplasia and thanatophoric dysplasia as ciliopathies. *Hum Mol Genet* 27: 1093-1105

Luo W, Ruba A, Takao D, Zweifel LP, Lim RYH, Verhey KJ, Yang W (2017) Axonemal Lumen Dominates Cytosolic Protein Diffusion inside the Primary Cilium. *Sci Rep* 7: 15793 Ou G, Blacque OE, Snow JJ, Leroux MR, Scholey JM (2005) Functional coordination of intraflagellar transport motors. *Nature* 436: 583-587

See SK, Hoogendoorn S, Chung AH, Ye F, Steinman JB, Sakata-Kato T, Miller RM, Cupido T, Zalyte R, Carter AP *et al* (2016) Cytoplasmic Dynein Antagonists with Improved Potency and Isoform Selectivity. *ACS Chem Biol* 11: 53-60

Williams CL, McIntyre JC, Norris SR, Jenkins PM, Zhang L, Pei Q, Verhey K, Martens JR (2014) Direct evidence for BBSome-associated intraflagellar transport reveals distinct properties of native mammalian cilia. *Nat Commun* 5: 5813

Yi P, Li WJ, Dong MQ, Ou G (2017) Dynein-Driven Retrograde Intraflagellar Transport Is Triphasic in *C. elegans* Sensory Cilia. *Curr Biol* 27: 1448-1461 e1447

Zhao C, Omori Y, Brodowska K, Kovach P, Malicki J (2012) Kinesin-2 family in vertebrate ciliogenesis. *Proceedings of the National Academy of Sciences* 109: 2388 - 2393

Zhou HM, Brust-Mascher I, Scholey JM (2001) Direct visualization of the movement of the monomeric axonal transport motor UNC-104 along neuronal processes in living *Caenorhabditis elegans*. *J Neurosci* 21: 3749-3755